# AVATAR: Learning to Align via Active Optimal Transport

Qi Yu [1]   Ruizhong Qiu [1]   Zhichen Zeng [1]   My T. Thai [2]   Huan Liu [3]   Hanghang Tong [1]

## Abstract

Alignment plays a fundamental role in many machine learning problems, such as multi-network analysis, multimodal learning, and point cloud registration. Recent works increasingly leverage optimal transport (OT) for distributional alignment, whose effectiveness largely depends on sparse supervision that is hard or costly to obtain in practice. Existing works, however, largely overlook how to actively acquire high-quality supervision to improve their alignment performance under OT frameworks. In this paper, we propose a principled a_ctive a_lignment framework for opt_imal t_ransport alignment called AVATAR. We quantify the informativeness of a candidate by measuring its gradient-based impact on the global alignment result, computed as the gradient propagation from the global alignment result to all possible supervisions of the candidate through the entropy-regularized OT formulation. While differentiating through OT is challenging given its constrained nature, we leverage the adjoint-state method to reformulate the computation to a linear system solvable by the conjugate gradient method with linear complexity and guaranteed convergence. By encoding the global alignment result via effective utility functions, AVATAR is applicable to general alignment problems under the OT framework. Extensive experiments on three representative alignment tasks demonstrate the effectiveness, scalability, and generalizability of the proposed AVATAR.

## 1. Introduction

Alignment is a critical steppingstone behind a wide range of machine learning problems, including but not limit to multi-network analysis (Du et al., 2021; Yan et al., 2022; Tang et al., 2023; Wang et al., 2023; Yu et al., 2025a; Zeng et al., 2023b; 2024b;a; 2025c), multimodal and cross-domain learning (Yilmaz et al., 2019; Chen et al., 2020; Cheng et al., 2022; Xu et al., 2024; Yoo et al., 2024; Ning et al., 2025), and point cloud registration. (Yu et al., 2021; 2023; Haitman et al., 2024). The general goal of these problems is to identify meaningful correspondence between two sets of data points, which facilitate various downstream machine learning tasks. For example, aligning nodes from different networks enables personalized recommendation across social platforms and helps fraud detection across transaction networks (Zeng et al., 2023a; Yu et al., 2025b). Aligning entities from different data modalities, e.g., image-text matching, enables automatic labeling of cross-modal data used for large-scale pre-training of multimodal foundation models (Han et al., 2021; Gan et al., 2022; Liang et al., 2024; Bartan et al., 2025; Wei et al., 2026).

Recently, optimal transport (OT) (Gabriel & Marco, 2019) has been increasingly adopted as an effective tool for solving alignment problems in general. By associating two sets of objects to be aligned (e.g., nodes from two networks) with two discrete probability distributions serving as the marginal constraints, OT-based alignment methods infer object-level alignment from the solved transport plan under carefully crafted cost function for specific tasks. Empowered by informative cost function and constrained optimization, OT-based methods naturally learn robust and deterministic alignment from a global view (Yu et al., 2025b;a), and have demonstrated remarkable performance across diverse alignment tasks (Chen et al., 2020; Qin et al., 2023; Yu et al., 2025b). Despite their success, Figure 1 shows that OT-based alignment approaches are sensitive to the quantity and quality of supervisions (Yu et al., 2025a), yet obtaining high-quality supervision is costly in practice (Zhou et al., 2021; Gan et al., 2022; Liu et al., 2024; Li et al., 2024). To date, few works have investigated how to actively acquire high-quality supervision in weakly supervised or unsupervised settings to effectively improve the performance of OT-based methods.

Although there exists sparse literature on active alignment, they bear the following three key limitations for OT-based methods. Firstly (*Limitation #1*), existing active alignment methods are not tailored to OT thus fail to utilize key components behind OT-based alignment, such as the cost function

[1]University of Illinois Urbana-Champaign [2]University of Florida [3]Arizona State University. Correspondence to: Hanghang Tong <htong@illinois.edu>.

*Proceedings of the 43rd International Conference on Machine Learning*, Seoul, South Korea. PMLR 306, 2026. Copyright 2026 by the author(s).

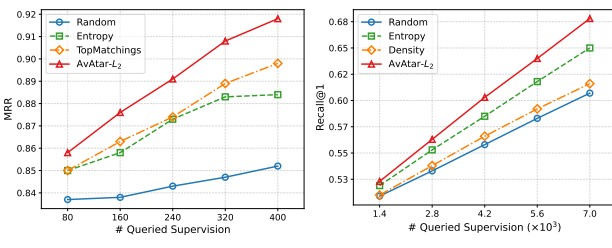

*(a)* Network Alignment    *(b)* Image-Text Grounding

*Figure 1.* **Sensitivity of active OT-based alignment methods w.r.t. the quality and quantity of supervision on network alignment and image-text grounding**. Observations: **(1. Quantity)** The performance of OT-based methods improve significantly by up to 15% with increased supervision level. **(2. Quality)** Under the same number of supervision, the performance of OT-based methods can differ significantly (up to 7%) by different query strategies, e.g., the proposed AVATAR-$L_2$ (red) vs. Random (blue).

and marginal constraints, which directly affect the alignment performance (Malmi et al., 2017; Cheng et al., 2019; Zhou et al., 2021). Secondly (*Limitation #2*), there lacks a principled method to quantify how newly acquired supervision would propagate through the OT formulation, making it difficult to assess the direct impact of a query on the alignment results (Malmi et al., 2017; Cheng et al., 2019). Thirdly (*Limitation #3*), prior efforts on active alignment mainly focus on designing task-specific query strategies (Ren et al., 2019; Zhou et al., 2021), such as active consistency-based network alignment methods, which are not readily generalizable to other alignment methods (e.g., OT-based methods) or alignment tasks (e.g., cross-domain alignment).

In this paper, we address these limitations by proposing a principled active alignment framework based on optimal transport, called AVATAR. AVATAR is designed intrinsically for OT-based alignment, which evaluates the informativeness of candidates by their posterior query impact on the global alignment results of OT, comprehensively utilizing key elements of OT to select the best candidates to query (*Limitation #1*). To quantify the posterior query impact of a candidate, AVATAR computes the gradient propagation from the global alignment result to all possible supervision signals of the candidate through the entropy-regularized OT formulation, capturing exactly how a new label would affect OT-based alignment through gradients (*Limitation #2*). However, a major challenge lies in differentiating through the OT formulation, since the transport plan is defined implicitly as the solution of a large-scale constrained optimization problem. To tackle this, we leverage the adjoint-state method to reformulate the computation of gradient-based impact to a linear system which can be solved efficiently via the conjugate gradient method with linear complexity and guaranteed convergence. By encoding the global alignment results via effective utility functions defined over the transport plan of OT, AVATAR is applicable to OT-based alignment methods across diverse alignment tasks with min-

imal modification (*Limitation #3*).

To validate the effectiveness of AVATAR, we conduct comprehensive experiments across three representative alignment tasks, including network alignment (NA) and two cross-domain alignment (CDA) tasks: image-text retrieval and image-text grounding. Extensive experiments covering 8 datasets, 4 OT-based alignment algorithms, and 9 baseline active learning methods demonstrate that AVATAR consistently outperform existing active learning approaches under the same query budget. We also show empirically that AVATAR achieves a good balance between alignment performance and efficiency, making it applicable to large-scale alignment problems.

Our main contributions are summarized as follows:

- **Problem.** To our best knowledge, we are the first to formalize the timely and important problem of active learning for OT-based alignment.
- **Method.** We propose a novel method AVATAR to quantify the informativeness of candidates by their impact on the alignment results, measured by gradient propagation through the OT formulation.
- **Analysis.** We provide extensive theoretical analysis of AVATAR to establish its correctness, linear time complexity, and linear convergence.
- **Evaluation.** Extensive experiments across diverse alignment tasks show that AVATAR consistently improves alignment performance and achieves a good balance between effectiveness and efficiency.

## 2. Preliminaries

In this section, we introduce preliminaries on optimal transport in Section 2.1, followed by OT-based alignment problem in Section 2.2. In this paper, we use bold uppercase letters for matrices (e.g., $\mathbf{T}$), bold lowercase letters for vectors (e.g., $\boldsymbol{\mu}$), calligraphic uppercase letters for sets (e.g., $\mathcal{X}$) and lowercase letters for scalars (e.g., $k$). Table 1 summarizes the main symbols used throughout the paper.

### 2.1. Optimal Transport

OT has emerged as a powerful mathematical tool for aligning two distributions (Santambrogio, 2015). Let $\boldsymbol{\mu} = \sum_{i=1}^{n} \mu_i \delta_{x_i}$ and $\boldsymbol{\nu} = \sum_{j=1}^{m} \nu_i \delta_{y_j}$ be two discrete probability distributions where $\delta$ denotes the Dirac measure. The discrete optimal transport problem seeks an optimal transport plan $\mathbf{T}$ that minimizes the total transport cost as follows:

$$\min_{\mathbf{T} \in \Pi(\boldsymbol{\mu}, \boldsymbol{\nu})} \langle \mathbf{C}, \mathbf{T} \rangle \quad (1)$$

where $\Pi(\boldsymbol{\mu}, \boldsymbol{\nu}) := \left\{ \mathbf{T} \in \mathbb{R}_+^{n \times m} | \mathbf{T}\mathbf{1}_m = \boldsymbol{\mu}, \mathbf{T}^\top \mathbf{1}_n = \boldsymbol{\nu} \right\}$, $\mathbf{C} \in \mathbb{R}_{\geq 0}^{n \times m}$ is the cost matrix where $\mathbf{C}_{i,j}$ measures the cost of transporting mass from the support point $x_i$ of $\boldsymbol{\mu}$ to point

$y_j$ of $\boldsymbol{\nu}$. The optimal value of Eq. (1) defines the *Wasserstein distance* between $\boldsymbol{\mu}$ and $\boldsymbol{\nu}$ under the cost matrix $\mathbf{C}$, and the resulting transport plan encodes the soft correspondence between points from the two distributions.

While Eq. (1) induces a linear programming problem of cubic complexity which is infeasible for large-scale applications, (Gabriel & Marco, 2019) introduces entropic regularization into Eq. (1) to approximate the original OT formulation:

$$\min_{\mathbf{T}\in\Pi(\boldsymbol{\mu},\boldsymbol{\nu})} \langle \mathbf{C}, \mathbf{T}\rangle - \epsilon\,\mathrm{Ent}(\mathbf{T}) \tag{2}$$

where $\mathrm{Ent}(\mathbf{T}) := -\sum_{i,j} \mathbf{T}_{i,j}(\log\mathbf{T}_{i,j} - 1)$ and $\epsilon > 0$ denotes the entropic regularization weight. Eq. (2) yields an $\epsilon$-strongly convex optimization problem that can be solved more efficiently with a quadratic complexity via the Sinkhorn algorithm (Nemirovski & Rothblum, 1999).

### 2.2. OT-based Alignment

A generalized definition for alignment problems based on OT can be summarized as follows (Zeng et al., 2023a; Yu et al., 2025b; Chen et al., 2020):

**Definition 2.1.** *OT-based Alignment*
**Given:** (1) two sets of objects $\mathcal{X} = \{x_i\}_{i=1}^n$ and $\mathcal{Y} = \{y_j\}_{j=1}^m$ to be aligned, and their associated marginal distributions $\boldsymbol{\mu}, \boldsymbol{\nu}$, (2) a cost function $\mathbf{C} \in \mathbb{R}^{n\times m}$, (3) an alignment supervision matrix $\mathbf{H} \in \{0,1\}^{n\times m}$ with $\mathbf{H}_{i,j} = 1$ indicating prior alignment between $x_i$ and $y_j$.
**Output:** an optimal transport plan $\mathbf{T}^* \in \Pi(\boldsymbol{\mu}, \boldsymbol{\nu})$ indicating the soft correspondence between $\mathcal{X}$ and $\mathcal{Y}$:

$$\mathbf{T}^* = \arg\min_{\mathbf{T}\in\Pi(\boldsymbol{\mu},\boldsymbol{\nu})} \langle \tilde{\mathbf{C}},\ \mathbf{T}\rangle,\ \tilde{\mathbf{C}} = (\mathbf{1}_{n\times m} - \beta\mathbf{H}) \odot \mathbf{C} \tag{3}$$

Eq. (3) follows the common practice (Zeng et al., 2023a; Yu et al., 2025b) of integrating supervisions into OT-based alignment by penalizing the cost entries of aligned pairs. $\beta$ in Eq. (3) denotes the penalizing factor. Under unsupervised settings, $\mathbf{H}$ becomes a zero matrix, making Eq. (3) equivalent to Eq. (1).

## 3. Problem Definition

We study the problem of active alignment based on optimal transport, whose goal is to maximally increase the performance of OT-based alignment methods by selectively acquiring a fixed amount of supervision from an oracle (e.g., a human annotator). We assume that the oracle can answer queries of the following form (Zhou et al., 2021):

*Given an object $x_s$ from the source set $\mathcal{X}$, which target object $y_t \in \mathcal{Y}$ is the correct alignment of $x_s$?*

Based on this, we formally define the active OT-based alignment problem as follows:

*Table 1.* Symbols and Notations.

| Symbol | Definition |
|---|---|
| $\mathcal{X}, \mathcal{Y}$ | object sets |
| $\boldsymbol{\mu}, \boldsymbol{\nu}$ | marginal distributions of OT |
| $\mathbf{C}$ | cost function of OT |
| $\mathbf{T}$ | transport map of OT |
| $\mathbf{H}$ | alignment supervision matrix |
| $f(\cdot), f(\mathbf{T})$ | alignment utility function defined over $\mathbf{T}$ |
| $\mathbf{1}$ | an all-one vector/matrix |
| $\mathbf{X}_{i,j}$ | entry of matrix $\mathbf{X}$ in row $i$ and column $j$ |
| $\odot$ | Hadamard product |
| $\langle\cdot,\cdot\rangle$ | inner product |

**Definition 3.1.** *Active OT-based Alignment*
**Given:** (1) two sets of objects $\mathcal{X} = \{x_i\}_{i=1}^n$ and $\mathcal{Y} = \{y_j\}_{j=1}^m$ to be aligned, (2) an OT-based alignment method with marginal distributions $\boldsymbol{\mu}, \boldsymbol{\nu}$, cost function $\mathbf{C}$, and solved transport plan $\mathbf{T}$, (3) an alignment supervision matrix $\mathbf{H}$ that encodes supervision signals, (4) a fixed query budget $k$, and (5) an oracle.
**Output:** a set of $k$ objects in $\mathcal{X}$ for the oracle to label the correct alignment in $\mathcal{Y}$, which maximally improves the alignment performance for the unlabeled objects in $\mathcal{X}$.

While the true alignment performance over unlabeled objects are typically unknown without access to test data, we quantify the quality of alignment results by effective utility functions introduced in Section 4.1.

Definition 3.1 is general and applicable to a wide range of alignment problems. Specifically, for *NA*, $\mathcal{X}$ and $\mathcal{Y}$ correspond to the node sets in the source and target networks; for *CDA*, $\mathcal{X}$ and $\mathcal{Y}$ denote instances from two different domains or modalities, e.g. image and text. This unified problem formulation allows us to define a task-agnostic active learning template for OT-based alignment methods applicable to diverse alignment tasks.

## 4. Methodology

In this section, we present the proposed active OT-based alignment framework AVATAR. We first introduce a quantitative gradient-based method for evaluating the informativeness of querying a source object, in Section 4.1. Then, we introduce the proposed AVATAR which leverages this gradient-based method for solving the active OT-based alignment problem, in Section 4.2. Finally, we provide theoretical analysis regarding the complexity and convergence of AVATAR, in Section 4.3.

### 4.1. Gradient-based Query Impact

A fundamental principle in active learning is to query the most *informative* objects for the learning task (Zhou et al.,

*Table 2.* Adopted utility functions $f(\mathbf{T})$ and their gradient w.r.t. the transport map $\mathbf{T}$. $\log(\cdot)$ denotes the element-wise logarithmic function, $\|\cdot\|_1$ is L1 norm, $\mathrm{tr}(\cdot)$ denotes the trace of a matrix, and $\mathbf{M}_1, \mathbf{M}_2$ are the graph Laplacian matrices of the two networks.

| Description | Function $f(\mathbf{T})$ | $\nabla_{\mathbf{T}} f(\mathbf{T})$ |
|---|---|---|
| Squared $L_2$ norm | $\|\mathbf{T}\|_2^2$ | $2\mathbf{T}$ |
| Entropy | $\|\mathbf{T} \odot \log(\mathbf{T})\|_1$ | $\log(\mathbf{T}) + \mathbf{1}_{n \times m}$ |
| Consistency | $\mathrm{tr}\left(\mathbf{T}^\top \mathbf{M}_1 \mathbf{T}\right) + \mathrm{tr}\left(\mathbf{T}\mathbf{M}_2\mathbf{T}^\top\right)$ | $2\mathbf{M}_1\mathbf{T} + 2\mathbf{T}\mathbf{M}_2$ |

2021; Li et al., 2024; Qiu & Tong, 2024), oftentimes via gradient-based impact estimation (He et al., 2024; Chen et al., 2024; 2026). In the context of alignment, it is thus desirable to quantify the informativeness of querying a candidate object through its potential impact on the overall alignment results (Zhou et al., 2021). Accordingly, we propose to evaluate the informativeness of querying a source object by its posterior impact on the global alignment results, encoded by an effective utility function $f(\mathbf{T}) : \mathbb{R}^{n \times m} \to \mathbb{R}$ defined over the transport plan $\mathbf{T}$. The utility function is designed to turn matrix $\mathbf{T}$ that indicates pairwise alignment into a scalar, quantifying the *global quality* of the alignment.

In this paper, we adopt two general-purpose utility functions for all alignment tasks, $f_{L_2}$ and $f_{\text{entropy}}$, and design an additional utility function $f_{\text{consist}}$ for the NA tasks. Definitions of different utility functions are listed in Table 2. We differentiate AVATAR of different utility functions by suffixes, e.g., AVATAR-$L_2$.

The intuition of $f_{L_2}$ and $f_{\text{entropy}}$ is straightforward: both functions encourage querying source objects whose true alignment leads to a more *deterministic* result $\mathbf{T}$ that approximates a permutation matrix when $n \approx m$ (Gabriel & Marco, 2019; Yu et al., 2025b). For the NA-specific function $f_{\text{consist}}$, it encourage the consistency principles widely adopted in NA (Zhang & Tong, 2016; Zeng et al., 2023a). Specifically, the first term $\mathrm{tr}(\mathbf{T}^\top \mathbf{M}_1 \mathbf{T}) = \frac{1}{2} \sum_{i,j} \mathbf{A}_{i,j}^{(1)} \|\mathbf{T}_{i,:} - \mathbf{T}_{j,:}\|_2^2$ with $\mathbf{A}^{(1)}$ denoting the graph adjacency matrix of the first network. It encourages alignment from the source to target network that contributes most to the consistency principle: the 1-hop neighbors of aligned nodes across different networks should share similar alignment results (i.e. corresponding rows in $\mathbf{T}$). Similarly, the second term encourages consistency when the alignment direction is reversed.

For the rest of this subsection, we first define the impact of a pairwise objects query on an utility function and introduce how to calculate this gradient-based impact. Then, we formally define the posterior impact of an individual object query on the utility function.

### 4.1.1. PAIRWISE OBJECTS QUERY IMPACT

**Definition 4.1.** *Pairwise Objects Query Impact*
Given a query of a pair of objects denoted as $p_{ij} = (x_i, y_j)$ where $x_i \in \mathcal{X}, y_j \in \mathcal{Y}$, and an utility function $f(\mathbf{T})$ defined over the transport plan $\mathbf{T}$, the query impact of $p_{ij}$ is defined

as the *gradient* of $f(\mathbf{T})$ w.r.t. the alignment supervision signal of this pair of objects, i.e., $\mathbf{H}_{i,j}$. Formally, the pairwise objects query impact is defined as $\mathcal{I}(p_{ij}) = \nabla_{\mathbf{H}_{i,j}} f$.

The calculation of $\mathcal{I}(p_{ij})$ for OT-based alignment method can be decomposed by the chain rule as follows

$$\mathcal{I}(p_{ij}) = \nabla_{\mathbf{H}_{i,j}} f = \langle \nabla_{\tilde{\mathbf{C}}} f, \nabla_{\mathbf{H}_{i,j}} \tilde{\mathbf{C}} \rangle \qquad (4)$$

The second term $\nabla_{\mathbf{H}_{i,j}} \tilde{\mathbf{C}}$ quantifies the impact of an added supervision to the supervised cost matrix $\tilde{\mathbf{C}}$. Following Eq. (3), $\nabla_{\mathbf{H}_{i,j}} \tilde{\mathbf{C}}$ can be computed easily as $\nabla_{\mathbf{H}_{i,j}} \tilde{\mathbf{C}} = -\beta \mathbf{C}_{i,j} \mathbf{E}$ where $\mathbf{E}_{i,j} = 1$ and 0 otherwise.

The main computation falls into the first term $\nabla_{\tilde{\mathbf{C}}} f$, which quantifies the impact of changing the supervised cost matrix $\tilde{\mathbf{C}}$ on the utility function $f(\mathbf{T})$. Computing $\nabla_{\tilde{\mathbf{C}}} f$ requires differentiating through the OT formulation, which is highly challenging for the following reason: as $\mathbf{T}$ is an implicit function w.r.t. $\tilde{\mathbf{C}}$ due to existence of marginal constraints, differentiating $\mathbf{T}$ w.r.t. $\tilde{\mathbf{C}}$, i.e., $\frac{d\mathbf{T}}{d\tilde{\mathbf{C}}}$, requires explicitly forming and inverting a Jacobian matrix of size $(nm)^2$, which would be computationally intractable at scale.

To tackle this computational challenge, we leverage the adjoint-state method to reformulate the computation of $\nabla_{\tilde{\mathbf{C}}} f$ to solving an adjoint linear system of size $(n + m)$.

**Lemma 4.2.** *The gradient of a utility function $f(\mathbf{T})$ w.r.t. the cost function $\tilde{\mathbf{C}}$ under the entropy-regularized OT formulation can be computed by solving a linear system with respect to adjoint vectors $\mathbf{y}_\alpha \in \mathbb{R}^n, \mathbf{y}_\beta \in \mathbb{R}^m$ as follows,*

$$\nabla_{\tilde{\mathbf{C}}} f = \frac{1}{\epsilon} \mathbf{T} \odot \left(\mathbf{y}_\alpha \mathbf{1}_m^\top + \mathbf{1}_n \mathbf{y}_\beta^\top - \nabla_{\mathbf{T}} f\right) \qquad (5)$$

*s.t.*

$$\underbrace{\begin{bmatrix} diag(\boldsymbol{\mu}) & \mathbf{T} \\ \mathbf{T}^\top & diag(\boldsymbol{\nu}) \end{bmatrix}}_{\mathbf{A}} \underbrace{\begin{bmatrix} \mathbf{y}_\alpha \\ \mathbf{y}_\beta \end{bmatrix}}_{\mathbf{y}} = \underbrace{\begin{bmatrix} (\mathbf{T} \odot \nabla_{\mathbf{T}} f) \mathbf{1}_m \\ (\mathbf{T} \odot \nabla_{\mathbf{T}} f)^\top \mathbf{1}_n \end{bmatrix}}_{\mathbf{b}} \qquad (6)$$

*where $\epsilon$ is the entropic regularization weight, $\boldsymbol{\mu}, \boldsymbol{\nu}$ are marginal distributions of OT, and $diag(\cdot)$ creates a diagonal matrix from a vector.*

The detailed proof of Lemma 4.2 can be found in Appendix A.1. The core idea is to implicitly differentiate the constrained optimality conditions of OT by the adjoint-state method, which reformulates the computation of $\nabla_{\tilde{\mathbf{C}}} f$ to solving a linear system (Sadr et al., 2024) without explicitly inverting a large Jacobian.

To solve the linear system, we show in Appendix A.2 that the coefficient matrix $\mathbf{A}$ of Eq. (6) is singular, making it impossible to directly solve Eq. (6) by matrix inversion. To address this issue, we resort to conjugate gradient (CG) method, which is summarized in the following lemma 4.3.

**Lemma 4.3.** *Eq.* (6) *can be solved via the conjugate gradient method with guaranteed convergence to global optimum.*

The detailed proof of Lemma 4.3 can be found in Appendix 4.3. In general, Eq. (6) renders a singular linear system solvable by the CG method, given that $\mathbf{A}$ is positive-semidefinite and $\mathbf{b}$ lies in the range of $\mathbf{A}$ in Eq. (6) (Kaass-chieter, 1988; Hayami, 2018). This linear system corresponding to a convex quadratic optimization problem, therefore CG is guaranteed to converge to the global optimum.

We present detailed analysis of the complexity and convergence rate of the CG method for Eq. (6) in Section 4.3, which shows that Eq. (6) can be solved with (1) a *linear* time complexity w.r.t. the number of objects in $\mathcal{X}$ and $\mathcal{Y}$, and (2) guaranteed convergence with a *linear* convergence rate. Combining Eq. (4)-(6) gives the final formulation of $\mathcal{I}(p_{ij})$, which quantifies the impact of querying a possible pairwise alignment $(x_i, y_j)$:

$$\mathcal{I}(p_{ij}) = -\frac{\beta}{\epsilon}\mathbf{C}_{i,j}\mathbf{T}_{i,j}\left(\mathbf{y}_\alpha\mathbf{1}_m^\top + \mathbf{1}_n\mathbf{y}_\beta^\top - \nabla_{\mathbf{T}}f\right)_{ij}$$

$$\text{s.t.} \begin{bmatrix} \text{diag}(\boldsymbol{\mu}) & \mathbf{T} \\ \mathbf{T}^\top & \text{diag}(\boldsymbol{\nu}) \end{bmatrix} \begin{bmatrix} \mathbf{y}_\alpha \\ \mathbf{y}_\beta \end{bmatrix} = \begin{bmatrix} (\mathbf{T} \odot \nabla_{\mathbf{T}}f)\mathbf{1}_m \\ (\mathbf{T} \odot \nabla_{\mathbf{T}}f)^\top\mathbf{1}_n \end{bmatrix}$$

$$(7)$$

### 4.1.2. POSTERIOR OBJECT QUERY IMPACT

**Definition 4.4.** *Posterior Object Query Impact*
For an object query denoted as $p_i = (x_i, \cdot)$ where $x_i \in \mathcal{X}$, the posterior query impact of $p_i$ on the utility function $f$ is defined as the aggregation of the pairwise objects query impact between object $x_i \in \mathcal{X}$ and all objects $y$ in the target set $\mathcal{Y}$, weighted by the corresponding rows of the transport plan $\mathbf{T}_{i,:}$ as a posterior. Formally, the posterior object query impact is defined as

$$\mathcal{I}(p_i) = \sum_{j=1}^{m} \mathbf{T}_{i,j}\mathcal{I}(p_{ij}) \qquad (8)$$

The intuition of using the transport plan $\mathbf{T}$ as posterior weights is straightforward: while it is unknown which target object will be the true alignment for a source object before the query, $\mathbf{T}_{i,j}$ encodes the posterior alignment probability between two objects $x_i \in \mathcal{X}$ and $y_j \in \mathcal{Y}$ conditioned on the observed data and supervision. Aggregating pairwise objects query impact weighted by $\mathbf{T}_{i,:}$ thus computes the *expected impact* of querying a source object $x_i$.

### 4.2. AVATAR

Based on Definition 4.1 and Definition 4.4, we propose a generic query algorithm for OT-based alignment as described in Algorithm 1, which allows customized utility function $f$ that quantifies the global alignment quality for different tasks. The key idea of AVATAR is to iteratively **(1)** select candidate objects with the largest posterior impact by

---

**Algorithm 1** AVATAR

1: **Input:** (1) two sets of objects $\mathcal{X}$ and $\mathcal{Y}$, (2) an OT-based alignment method with marginal distributions $\boldsymbol{\mu}, \boldsymbol{\nu}$ and cost function $\mathbf{C}$, (3) an alignment supervision matrix $\mathbf{H}$, (4) a total query budget $k$, (5) query batch size $n_b$, (6) a query pool $\mathcal{P} \subseteq \mathcal{X}$, (7) an oracle, (8) an utility function $f$ for encoding the alignment result.
2: **Output:** (1) a set of $k$ objects $\mathcal{Q} \subseteq \mathcal{X}$ for query, (2) the alignment matrix $\mathbf{T}$ between $\mathcal{X}$ and $\mathcal{Y}$.
3: Initialize the query set $\mathcal{Q} = \emptyset$;
4: Remove source objects whose alignment are known from the query pool, i.e.,
$\mathcal{P} \leftarrow \mathcal{P} \backslash \left\{ x_i \in \mathcal{X} \mid \sum_j \mathbf{H}_{i,j} > 0 \right\}$;
5: Compute the current alignment matrix $\mathbf{T}$ by the input OT-based method via entropy-regularized OT solver;
6: **while** $|\mathcal{Q}| < k$ **do**
7:     Compute the posterior query impact of all source objects $\mathcal{I}$ using Eq. (8);
8:     Initialize batch query set $\mathcal{Q}_b = \emptyset$;
9:     **while** $|\mathcal{Q}_b| < n_b$ **do**
10:        Update $\mathcal{Q}_b \leftarrow \mathcal{Q}_b \cup \{x^* = \arg\max_{x_i \in \mathcal{P}} \mathcal{I}(p_i)\}$;
11:        Update $\mathcal{P} \leftarrow \mathcal{P} \backslash \{x^*\}$;
12:    **end while**
13:    Query for the correct alignment of objects in $\mathcal{Q}_b$;
14:    Update the alignment supervision matrix $\mathbf{H}$;
15:    Re-compute $\mathbf{T}$ by the OT-based alignment method;
16:    Update $\mathcal{Q} = \mathcal{Q} \cup \mathcal{Q}_b$;
17: **end while**
18: **return** $\mathcal{Q}$ and $\mathbf{T}$;

---

Eq. (8) (Steps 7 and 10); **(2)** query the oracle for their true alignment (Step 13); **(3)** update the alignment supervision matrix $\mathbf{H}$ accordingly (Step 14); and **(4)** re-compute the alignment results by an OT-based method (Step 15).

AVATAR operates under both weakly supervised and unsupervised settings by different initializations of the alignment supervision matrix $\mathbf{H}$. For weakly supervised alignment tasks where partial ground-truth alignment is pre-known, $\mathbf{H}_{i,j}$ is set to 1 if object $x_i$ and $y_j$ are known to be aligned, and 0 otherwise; for unsupervised alignment tasks, $\mathbf{H}_{i,j}$ is set to be a zero matrix. Similarly, when updating $\mathbf{H}$ during the query process, we set the corresponding entries of queried alignment pairs in $\mathbf{H}$ to 1. While we adopt a binary supervision matrix $\mathbf{H}$, AVATAR can be naturally extended to a soft supervision matrix $\mathbf{H}$ with continuous values.

### 4.3. Theoretical Analysis

In this subsection, we provide theoretical analysis regarding the complexity and convergence rate of proposed AVATAR.

**Theorem 4.5.** *(Time & Space Complexity of* AVATAR-$L_2$/ENTROPY/CONSIST*)*
*The time complexity is* $\mathcal{O}\left(\frac{k}{n_b}K(n+m)\right)$ *for* AVATAR-

$L_2$/ENTROPY, and $\mathcal{O}\left(\frac{k}{n_b}(K(n+m)+e)\right)$ for AVATAR-CONSIST, where $e$ denotes the number of edges in the networks[1]. The space complexity of AVATAR-$L_2$/ENTROPY/CONSIST is $\mathcal{O}(nm)$. $k$ is the total query budget, $n_b$ is the batch query size, $K$ is the number of iterations of the CG method, and $n, m$ are number of objects in the source and target sets, respectively.

The detailed proof of Theorem 4.5 can be found in Appendix A.4. In general, we reduce the complexity of AVATAR-$L_2$/ENTROPY/CONSIST to linear by utilizing the *sparsity* of the transport plan $\mathbf{T}$ of OT-based method, as visualized in Figure 5. Note that while the total complexity of AVATAR may depend on the selected utility function $f$, the coefficient martix $\mathbf{A}$ in Eq. (6) is agnostic to $f$ thus the CG method can always leverage the sparsity of $\mathbf{T}$ for OT-based alignment to approximate a linear time complexity of $\mathcal{O}(K(n+m))$ w.r.t. the number of objects in $\mathcal{X}$ and $\mathcal{Y}$.

**Theorem 4.6.** *(Convergence Rate of* AVATAR*)*
*The conjugate gradient method applied to the linear system of Eq.* (6) *converges at a linear convergence rate of* $\frac{\sqrt{\lambda_1/\lambda_r}-1}{\sqrt{\lambda_1/\lambda_r}+1}$*, where* $\lambda_1, \lambda_r$ *denotes the largest/smallest nonzero eigenvalues of* $\mathbf{A}$ *in Eq.* (6)*.*

The detailed proof of Theorem 4.6 can be found in Appendix A.5. This linear convergence rate guarantees fast and stable convergence of the CG method, making the gradient computation in AVATAR efficient and scalable.

# 5. Experiments

In this section, we carry out comprehensive experiments and analyses to evaluate the proposed AVATAR from the following aspects:

- **Q1.** How effective is the proposed AVATAR across different alignment problems?
- **Q2.** How efficient and scalable is AVATAR?
- **Q3.** How is the empirical convergence of AVATAR?
- **Q4.** How sensitive is the proposed AVATAR to different parameters and design choices?

## 5.1. Experimental Setup

We benchmark the proposed AVATAR on three different alignment tasks, including network alignment (NA), image-text retrieval, and image-text grounding.

**Network Alignment.** NA aims to find node-level correspondence across different networks. For *datasets*, we adopt 4 real-world datasets Phone-Email (Zhang et al., 2017), ACM-DBLP-P (Tang et al., 2008), Douban (Zhang

& Tong, 2016), and ACM-DBLP-A (Tang et al., 2008), covering both plain and attributed networks. For *baseline OT-based NA methods*, we adopt PARROT (Zeng et al., 2023a) and JOENA (Yu et al., 2025b). For *baseline query strategies*, we adopt RANDOM, ENTROPY, MARGIN (Li et al., 2024), BETWEENNESS (Macskassy, 2009), CONTRASTIVE, GIBBSMATCHINGS (Malmi et al., 2017), and TOPMATCHINGS (Malmi et al., 2017). We report the mean reciprocal rank (MRR) as the benchmarking metric for NA.

**Image-Text Retrieval.** Image-text retrieval aims to retrieve the most relevant text given an image query, or vice versa. In this paper, we focus on the former setting. For *datasets*, we adopt 2 widely used datasets CIFAR-10-C and ImageNet-C (Hendrycks & Dietterich, 2019). For *baseline OT-based CDA methods*, we adopt the Wasserstein (W) and Fused Gromov-Wasserstein (FGW) variants of GOT (Chen et al., 2020). For *baseline query strategies*, we adopt RANDOM, ENTROPY, MARGIN, LEAST CONFIDENT, DENSITY, and DIVERSITY (Li et al., 2024). We report the Recall@1 as the benchmarking metric for image-text retrieval.

**Image-Text Grounding.** Image-text grounding aims to identify the fine-grained correspondence between phrases in a sentence and objects (or regions) in an image (Li et al., 2022). For *datasets*, we adopt 2 widely used datasets COCO (Lin et al., 2014) and Flickr30K Entities (Plummer et al., 2015). We adopt the same baseline OT-based alignment methods, query strategies, and evaluation metric as that of image-text retrieval.

Detailed experimental settings, as well as descriptions of datasets and introduction for different query strategies, are included in Appendix B.

## 5.2. Benchmarking Results

### 5.2.1. NETWORK ALIGNMENT

For NA tasks, we adopt the task-agonistic AVATAR-$L_2$, and AVATAR-CONSIST which is designed specifically for NA. We compare these two versions of the proposed method with other baseline query methods in alignment performance measured by MRR. The results are summarized in Table 3. We observe that **(1) AVATAR-$L_2$/CONSIST achieve state-of-the-art performance across all datasets.** They consistently outperform all baselines by up to 2.8% in MRR, demonstrating the effectiveness of AVATAR on both plain and attributed NA tasks. **(2) Existing active NA methods becomes less effective on OT-based algorithms.** We notice that GIBBSMATCHINGS and TOPMATCHINGS designed specifically for active NA are consistently outperformed by general-purpose active query strategies, e.g., ENTROPY, when applied to OT-based methods. This is because the deterministic transport plan gives consistent sampling results, making it difficult for GIBBSMATCHINGS and TOPMATCHINGS to characterize the uncertainty of a candidate. **(3) AVATAR**

---

[1]Without loss of generality, we assume $\mathcal{O}(e) \approx \mathcal{O}(e_1) \approx \mathcal{O}(e_2)$ where $e_i$ denotes the number of edges in the $i$-th networks (Zeng et al., 2023a; Yu et al., 2025b).

*Table 3.* Benchmarking results on network alignment in MRR. The **1st**/2nd best results are highlighted in **bold** and underline, respectively.

| Dataset | Phone-Email | | ACM-DBLP-P | | Douban | | ACM-DBLP-A | |
|---|---|---|---|---|---|---|---|---|
| #-th Query Round | 5 | 10 | 5 | 10 | 5 | 10 | 5 | 10 |
| PARROT (Zeng et al., 2023a) | | | | | | | | |
| RANDOM | 0.517 | 0.568 | 0.723 | 0.740 | 0.730 | 0.751 | 0.815 | 0.829 |
| ENTROPY | 0.533 | 0.615 | 0.774 | 0.840 | 0.765 | 0.791 | 0.863 | 0.926 |
| MARGIN | 0.538 | 0.618 | 0.766 | 0.824 | 0.756 | 0.819 | 0.862 | 0.915 |
| BETWEENNESS | 0.530 | 0.599 | 0.762 | 0.811 | 0.740 | 0.780 | 0.852 | 0.889 |
| DENSITY | 0.533 | 0.610 | 0.729 | 0.770 | 0.739 | 0.761 | 0.834 | 0.854 |
| GIBBSMATCHINGS | 0.517 | 0.576 | 0.717 | 0.736 | 0.736 | 0.756 | 0.810 | 0.824 |
| TOPMATCHINGS | 0.512 | 0.576 | 0.752 | 0.760 | 0.761 | 0.817 | 0.831 | 0.846 |
| AVATAR-$L_2$ | 0.539 | 0.629 | **0.787** | **0.858** | **0.782** | **0.837** | **0.876** | **0.936** |
| AVATAR-CONSIST | **0.544** | **0.630** | 0.785 | 0.856 | 0.779 | 0.835 | **0.876** | 0.935 |
| JOENA (Yu et al., 2025b) | | | | | | | | |
| RANDOM | 0.582 | 0.648 | 0.760 | 0.786 | 0.839 | 0.852 | 0.821 | 0.837 |
| ENTROPY | 0.680 | 0.778 | 0.829 | 0.911 | 0.865 | 0.884 | 0.910 | 0.972 |
| MARGIN | 0.652 | 0.762 | 0.816 | 0.915 | 0.868 | 0.915 | 0.905 | 0.971 |
| BETWEENNESS | 0.642 | 0.730 | 0.824 | 0.892 | 0.855 | 0.872 | 0.883 | 0.926 |
| DENSITY | 0.561 | 0.639 | 0.794 | 0.801 | 0.866 | 0.885 | 0.838 | 0.861 |
| GIBBSMATCHINGS | 0.589 | 0.661 | 0.764 | 0.792 | 0.840 | 0.848 | 0.828 | 0.845 |
| TOPMATCHINGS | 0.589 | 0.667 | 0.782 | 0.809 | 0.870 | 0.898 | 0.839 | 0.859 |
| AVATAR-$L_2$ | 0.682 | 0.800 | 0.845 | 0.927 | **0.883** | **0.918** | **0.924** | **0.981** |
| AVATAR-CONSIST | **0.691** | **0.806** | **0.850** | **0.929** | 0.879 | 0.915 | **0.924** | 0.980 |

*Table 4.* Benchmarking results on image-text retrieval in Recall@1. The **1st**/2nd best results are highlighted in **bold** and underline, respectively.

| Algorithm | GOT-W (Chen et al., 2020) | | | | GOT-FGW (Chen et al., 2020) | | | |
|---|---|---|---|---|---|---|---|---|
| Dataset | CIFAR-10-C | | ImageNet-C | | CIFAR-10-C | | ImageNet-C | |
| #-th Query Round | 5 | 10 | 5 | 10 | 5 | 10 | 5 | 10 |
| RANDOM | 0.564 | 0.607 | 0.374 | 0.454 | 0.534 | 0.585 | 0.322 | 0.392 |
| ENTROPY | 0.583 | 0.641 | 0.394 | 0.489 | 0.546 | 0.610 | 0.328 | 0.404 |
| MARGIN | 0.574 | 0.622 | 0.393 | 0.490 | 0.547 | 0.610 | 0.331 | 0.411 |
| LEAST CONFIDENT | 0.579 | 0.632 | 0.397 | 0.496 | 0.548 | 0.613 | 0.334 | 0.414 |
| DENSITY | 0.571 | 0.614 | 0.381 | 0.464 | 0.542 | 0.602 | 0.327 | 0.401 |
| DIVERSITY | 0.568 | 0.622 | 0.396 | 0.494 | 0.533 | 0.597 | 0.328 | 0.404 |
| AVATAR-ENTROPY | **0.586** | **0.648** | **0.402** | **0.509** | **0.553** | **0.619** | **0.342** | **0.430** |
| AVATAR-$L_2$ | 0.584 | 0.645 | 0.399 | 0.505 | 0.551 | 0.618 | 0.340 | 0.426 |

**benefits from effective utility functions tailored for specific alignment tasks.** On plain networks (Phone-Email and ACM-DBLP-P), AVATAR-CONSIST designed specifically for NA tasks typically outperforms AVATAR-$L_2$ with up to 0.6% improvement in MRR, indicating that consistency principles are particularly useful when attribute information is missing. In contrast, AVATAR-$L_2$ generally achieves better performance than AVATAR-CONSIST on attributed networks (Douban, ACM-DBLP-A), suggesting that deterministic alignment results given by $f_{L_2}$, as mentioned in Section 4, are more helpful when informative attributes are available. Complete results that shows MRR vs. query round for all baselines across three different alignment tasks can be found in Appendix C.1.

### 5.2.2. CROSS-DOMAIN ALIGNMENT

We include two tasks for benchmarking AVATAR on CDA: *image-text retrieval* and *image-text grounding*. For both CDA tasks, we compare AVATAR-$L_2$/ENTROPY with other baseline query methods in alignment performance by Recall@1. The results of image-text retrieval and image-text grounding can be found in Tables 4 and 5, respectively.

*Table 5.* Benchmarking results on image-text grounding in Recall@1. The **1st**/2nd best results are highlighted in **bold** and underline, respectively.

| Algorithm | GOT-W (Chen et al., 2020) | | | | GOT-FGW (Chen et al., 2020) | | | |
|---|---|---|---|---|---|---|---|---|
| Dataset | COCO | | Flickr30K | | COCO | | Flickr30K | |
| #-th Query Round | 5 | 10 | 5 | 10 | 5 | 10 | 5 | 10 |
| RANDOM | 0.510 | 0.577 | 0.356 | 0.441 | 0.545 | 0.607 | 0.550 | 0.628 |
| ENTROPY | 0.534 | 0.620 | 0.363 | 0.455 | 0.569 | 0.650 | 0.563 | 0.652 |
| MARGIN | 0.532 | 0.625 | 0.359 | 0.449 | 0.568 | 0.656 | 0.562 | 0.651 |
| LEAST CONFIDENT | 0.535 | 0.625 | 0.360 | 0.448 | 0.570 | 0.655 | 0.567 | 0.657 |
| DENSITY | 0.518 | 0.585 | 0.355 | 0.439 | 0.553 | 0.616 | 0.567 | 0.657 |
| DIVERSITY | 0.533 | 0.626 | 0.346 | 0.426 | 0.568 | 0.656 | 0.561 | 0.650 |
| AVATAR-ENTROPY | **0.538** | **0.628** | **0.366** | **0.461** | 0.578 | 0.668 | 0.573 | 0.670 |
| AVATAR-$L_2$ | 0.538 | 0.627 | 0.365 | 0.459 | **0.582** | **0.678** | **0.575** | **0.671** |

*Figure 2.* MRR vs. total query time of AVATAR and baselines, showing that AVATAR achieves up to **25× speed-up and 8.1% improvement in MRR** compared to existing active NA methods.

We observe that **(1) AVATAR-ENTROPY/$L_2$ consistently achieve state-of-the-art performance on both CDA tasks**, demonstrating the effectiveness of AVATAR on OT-based alignment methods for CDA. **(2) AVATAR-ENTROPY aligns well with the objective of entropic OT**. AVATAR-ENTROPY typically outperforms AVATAR-$L_2$ under most settings except for the image-text grounding task by GOT-FGW, which implies that $f_{entropy}$ generally aligns better with the objective of entropic OT alignment. **(3) Gradients characterize the informativeness of a candidate better than the plain transport plan T**. AVATAR-ENTROPY consistently outperforms ENTROPY by up to 2.6% in Recall@1, suggesting that the gradients propagated through OT provide more accurate measures of the informativeness of candidates than directly applying the same utility function on **T**.

### 5.3. Efficiency Results

We study the effectiveness-efficiency trade-off of proposed AVATAR-$L_2$/CONSIST compared with other baseline query methods on phone-email and Douban datasets using PARROT (Zeng et al., 2023a). The MRR vs. query time trade-off results are shown in Figure 2. We can see that **AVATAR-$L_2$/CONSIST achieve a good balance between the alignment performance and query time**. Specifically, AVATAR achieves up to 25× speed-up compared to TOPMATCHINGS, and up to 8.1% improvement in MRR compared to GIBBSMATCHINGS, demonstrating both the effectiveness and efficiency of AVATAR over existing active NA methods. While most other baselines are slightly faster, they are consistently less effective in improving the alignment

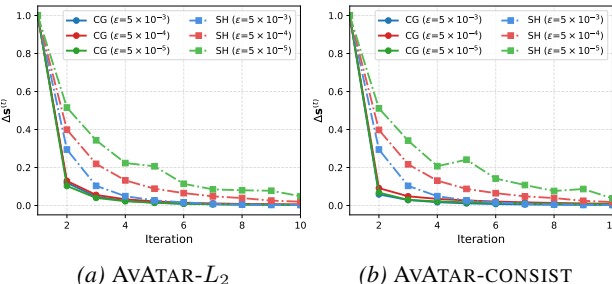

*(a)* AVATAR-$L_2$          *(b)* AVATAR-CONSIST

*Figure 3.* Convergence analysis of the conjugate gradient (CG) and the Sinkhorn (SH) algorithm in AVATAR under different entropic regularization weight $\epsilon$.

performance than AVATAR. More results on scalability are included in Appendix C.2.

## 5.4. Convergence Results

We evaluate the convergence of the the conjugate gradient (CG) method for solving Eq. (6) and that of the Sinkhorn (SH) algorithm used for solving OT-based alignment, in terms of the difference between two consecutive solutions $\Delta \mathbf{s}^{(t)} = \|\mathbf{s}^{(t)} - \mathbf{s}^{(t-1)}\|_1$, on Phone-Email. For CG, $\mathbf{s} = \mathbf{y}$ in Eq. (6); for SH, $\mathbf{s} = \text{vec}(\mathbf{T})$. Results are normalized to ensure a fair comparison. The corresponding convergence result of AVATAR-$L_2$ and AVATAR-CONSIST, are shown in Figure 3. We observe that **(1) CG converges empirically** when applied to Eq. (6), which validates the convergence of AVATAR. **(2) The convergence of SH is more sensitive to $\epsilon$ than CG.** As the entropic regularization weight $\epsilon$ decreases, SH becomes increasingly ill-conditioned and harder to converge, while the convergence rate of CG remains relatively stable. **(3) CG converges faster than SH under the same** $\epsilon$, suggesting that the CG method, with a linear time complexity per iteration, incurs *negligible overhead* compared to the Sinkhorn algorithm for solving OT.

## 5.5. Further Studies

### 5.5.1. HYPERPARAMETER ANALYSIS

**Query Budget $k$.** The impact of the query budget $k$ is presented in Tables 3-5 and Appendix C.1, which shows that the alignment performance of AVATAR improves monotonically as the query budget $k$ increases.

**Query Batch Size $n_b$.** We perform a sensitivity study on the query batch size $n_b$ in Figure 4. We adopt AVATAR-$L_2$ and test the performance of PARROT on two datasets as the query batch size $n_b$ changes under different query budget $k$. We can see in Figure 4 that the performance of AVATAR-$L_2$ remains stable under different choices of $n_b$, with only slight performance improvement when $n_b$ is smaller on Douban.

### 5.5.2. ABLATION STUDY

**Sparse vs. Dense Matrix Operation in CG.** As mentioned in Section 4.3, AVATAR leverages the sparsity of the transport plan $\mathbf{T}$ to achieve linear time complexity. To

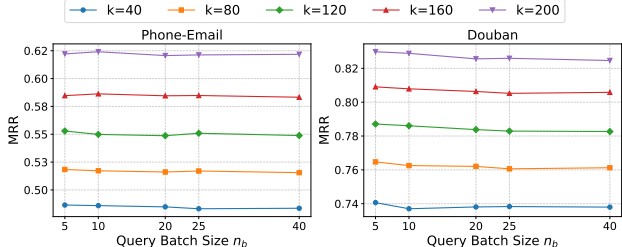

*Figure 4.* Parameter study for query batch size $n_b$ on AVATAR-$L_2$.

verify the effectiveness and efficiency of sparse matrix operations in AVATAR, we compare the alignment performance of AVATAR with AVATAR(Dense), which computes Eq. (7) and Eq. (8) by dense matrix operations. Specifically, we compare AVATAR-$L_2$ with AVATAR-$L_2$ (Dense), and AVATAR-CONSIST with AVATAR-CONSIST (Dense). The results on two datasets are shown in Table 6. We observe that (1) **Sparse matrix operations do not harm the empirical performance of AVATAR**, as we can see that the alignment performance of AVATAR and AVATAR(Dense) remains close across different datasets. **(2) Sparse matrix operation significantly improves the efficiency of AVATAR**. AVATAR runs significantly faster than AVATAR(Dense) with up to $1010/118 \approx 8.6$ times speed-up, demonstrating the efficiency of sparse matrix operations on AVATAR.

*Table 6.* Ablation study on sparse matrix operations adopted by AVATAR.

| Dataset | Douban | | ACM-DBLP-A | |
|---|---|---|---|---|
| Metric | MRR | Time(s) | MRR | Time(s) |
| AVATAR-$L_2$(Dense) | **0.839** | 19 | 0.936 | 879 |
| AVATAR-$L_2$ | 0.837 | **5.1** | 0.936 | **107** |
| AVATAR-CONSIST(Dense) | 0.834 | 20 | **0.936** | 1010 |
| AVATAR-CONSIST | **0.835** | **5.4** | 0.935 | **118** |

**Posterior vs. Uniform Aggregation.** We verify the necessity of using transport plan $\mathbf{T}$ as a posterior in Eq. (8) by comparing the performance of AVATAR with AVATAR(Uniform), which aggregated pairwise objects query impact in Eq. (4) uniformly for computing the object-level query impact, i.e., $\mathcal{I}(p_i) = \sum_{j=1}^{m} \mathcal{I}(p_{ij})$. As shown in Table 7, AVATAR consistently outperforms AVATAR(Uniform) on both Douban and ACM-DBLP-A under different choices of utility functions, demonstrating the informativeness of the transport plan $\mathbf{T}$ as a posterior.

## 5.6. Visualizations

**Sparsity of Transport Plans** To verify the sparsity of the output transport plan $\mathbf{T}$, we visualize randomly sampled $\mathbf{T}$ (300x300) of different OT-based alignment methods under their default hyperparameter settings in Figure 5. The results show that **the T returned by different OT-based methods are empirically sparse** across different alignment tasks, **with over 99% entries close to zero**, suggesting that

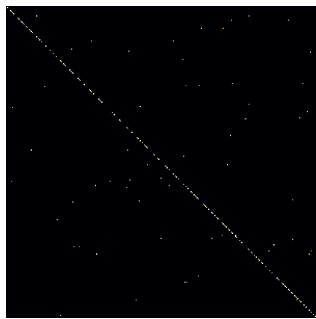 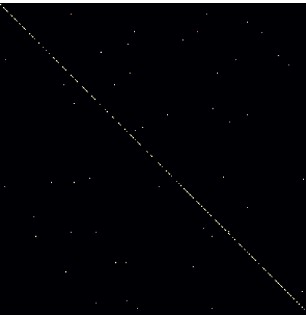 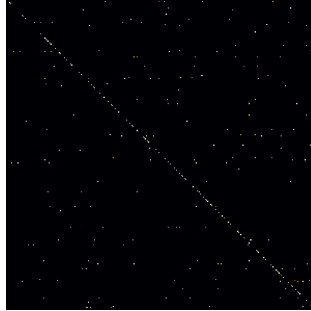 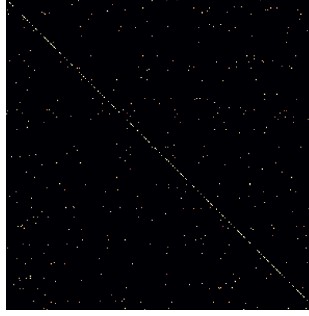

*(a)* PARROT on Phone-Email       *(b)* JOENA on Phone-Email       *(c)* GOT-W on ImageNet-C       *(d)* GOT-W on COCO

*Figure 5.* Visualization of transport plans **T** of different OT-based methods on different datasets under default hyperparameter settings in their original papers. Lighter pixels denotes higher value. Ground-truth alignment are moved to diagonals for better visualization.

*Table 7.* Ablation study on posterior aggregation of pairwise objects query impact in AVATAR.

| Dataset | Douban | | Phone-Email | |
|---|---|---|---|---|
| Metric | Hits@1 | MRR | Hits@1 | MRR |
| AVATAR-$L_2$(Uniform) | 0.558 | 0.669 | 0.397 | 0.522 |
| AVATAR-$L_2$ | **0.755** | **0.837** | **0.495** | **0.629** |
| AVATAR-CONSIST(Uniform) | 0.572 | 0.677 | 0.379 | 0.506 |
| AVATAR-CONSIST | **0.753** | **0.835** | **0.498** | **0.630** |

OT-based methods typically choose a small entropic regularization weight $\epsilon$ to approximate deterministic, one-to-one alignment, making it possible for AVATAR to achieve linear time complexity by leveraging the sparsity of **T**.

**Drifts of Transport Plans.** To improve interpretability regarding the corrective benefit of the oracle, we include an additional study on the drift of $\mathbf{T}^{(i)}$ from $\mathbf{T}^{(0)}$ in the $i$-th query round, measured by the Frobenius norm of their difference (i.e., $\|\mathbf{T}^{(i)} - \mathbf{T}^{(0)}\|_{\mathrm{F}}$. The results are shown in Figure 11, which shows that **(1)** $\mathbf{T}^{(i)}$ **gradually drifts from** $\mathbf{T}^{(0)}$ **with increasing level of supervision along query rounds**, and **(2) the drift of T positively correlates with the performance improvement of OT-based alignment methods**, suggesting that the acquired supervision from the oracle gradually corrects **T** to infer accurate alignment.

## 6. Related Work

### 6.1. Optimal Transport

OT has recently attracted increasing attention in alignment problems for its effectiveness in aligning distributional structures and its efficiency enabled by entropic regularization. For NA, PARROT (Zeng et al., 2023a) proposes a regularized OT-based algorithm and integrates consistency principles for effective cost design. JOENA (Yu et al., 2025b) further unifies embedding learning and OT optimization, enabling end-to-end training of an effective and robust NA model. For CDA, GOT (Chen et al., 2020) combines both Wasserstein and Gromov-Wasserstein distance and proposes a principled framework for regularizing cross-domain align-

ment. DCOT (Wang et al., 2024) designs a dual-view OT framework for cross-modality retrieval. While OT has been used extensively for alignment, few works have studied OT-based alignment methods under active learning settings.

### 6.2. Active Alignment

Active alignment falls into the broad category of active learning, which studies how to maximally improve the performance of a machine learning model by querying as few supervision as possible from an oracle (Ren et al., 2021; Li et al., 2024). The sparse literature on active alignment mainly focus on network data. (Malmi et al., 2017) proposes two active NA methods that samples the most uncertain nodes via bipartite matching. Utilizing inter-network meta diagram and link selection, ActiveIter (Ren et al., 2019) introduces an active NA methods specifically for social networks. Attent (Zhou et al., 2021) proposes an active learning method for attributed consistency-based NA. While promising for specific NA tasks, they rely on network data and are not readily applicable to other alignment tasks such as cross-domain alignment, which are critical stepping stones for cross-domain and multi-modal machine learning (Gan et al., 2022; Liang et al., 2024; Wei et al., 2024; Zeng et al., 2025b; 2026b; 2025a; 2026a).

## 7. Conclusion

In this paper, we study the active alignment problem based on optimal transport. We select the most informative candidates to query based on their impact on the global alignment results encoded by effective utility functions, computed by gradient propagation through the entropy-regularized OT formulation. To differentiate through OT, we leverage the adjoint-state method to reformulate the computation to solving a linear system, which is solvable via the conjugate gradient method with linear complexity and guaranteed convergence. Based on these, we propose a generic active learning framework AVATAR for OT-based alignment applicable to diverse alignment tasks. Extensive experiments across three different alignment tasks demonstrate the effectiveness and scalability of AVATAR on real-world datasets.

## Impact Statement

This paper presents work whose goal is to advance the field of Active Machine Learning and Optimal Transport. There are many potential societal consequences of our work, none of which we feel must be specifically highlighted here.

## Acknowledgment

This work is supported by NSF (2433308, 2433309, 2416606) and AFOSR (FA9550-24-1-0002). The content of the information in this document does not necessarily reflect the position or the policy of the Government, and no official endorsement should be inferred. The U.S. Government is authorized to reproduce and distribute reprints for Government purposes notwithstanding any copyright notation here on.

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

# A. Proof

## A.1. Proof of Lemma 4.2

**Lemma.** *The gradient of a utility function $f(\mathbf{T})$ w.r.t. the cost function $\tilde{\mathbf{C}}$ under the entropy-regularized OT formulation can be computed by solving a linear system with respect to adjoint vectors $\mathbf{y}_\alpha \in \mathbb{R}^n, \mathbf{y}_\beta \in \mathbb{R}^m$ as follows,*

$$\nabla_{\tilde{\mathbf{C}}} f = \frac{1}{\epsilon} \mathbf{T} \odot \left( \mathbf{y}_\alpha \mathbf{1}_m^\top + \mathbf{1}_n \mathbf{y}_\beta^\top - \nabla_{\mathbf{T}} f \right) \tag{9}$$

*s.t.*

$$\underbrace{\begin{bmatrix} diag(\boldsymbol{\mu}) & \mathbf{T} \\ \mathbf{T}^\top & diag(\boldsymbol{\nu}) \end{bmatrix}}_{\mathbf{A}} \underbrace{\begin{bmatrix} \mathbf{y}_\alpha \\ \mathbf{y}_\beta \end{bmatrix}}_{\mathbf{y}} = \underbrace{\begin{bmatrix} \left( \mathbf{T} \odot \nabla_{\mathbf{T}} f \right) \mathbf{1}_m \\ \left( \mathbf{T} \odot \nabla_{\mathbf{T}} f \right)^\top \mathbf{1}_n \end{bmatrix}}_{\mathbf{b}} \tag{10}$$

*where $\epsilon$ is the entropic regularization weight, $\boldsymbol{\mu}, \boldsymbol{\nu}$ are marginal distributions of OT, and $diag(\cdot)$ creates a diagonal matrix from a vector.*

*Proof.* Consider the entropy-regularized OT formulation in Eq. (2). We introduce two dual variables $\boldsymbol{\alpha} \in \mathbb{R}^n, \boldsymbol{\beta} \in \mathbb{R}^m$ for the marginal constraints of OT, and gives the Lagrangian of Eq. (2) as

$$\mathcal{L}(\mathbf{T}, \boldsymbol{\alpha}, \boldsymbol{\beta}) = \langle \tilde{\mathbf{C}}, \mathbf{T} \rangle + \epsilon \sum_{i,j} \mathbf{T}_{i,j} \left( \log \mathbf{T}_{i,j} - 1 \right) + \boldsymbol{\alpha}^\top \left( \boldsymbol{\mu} - \mathbf{T} \mathbf{1}_m \right) + \boldsymbol{\beta}^\top \left( \boldsymbol{\nu} - \mathbf{T}^\top \mathbf{1}_n \right) \tag{11}$$

The first order condition gives

$$\frac{\partial \mathcal{L}(\mathbf{T}, \boldsymbol{\alpha}, \boldsymbol{\beta})}{\partial \mathbf{T}_{i,j}} = \tilde{\mathbf{C}}_{i,j} + \epsilon \log \mathbf{T}_{i,j} - \boldsymbol{\alpha}_i - \boldsymbol{\beta}_j = 0 \tag{12}$$

which yields

$$\mathbf{T}_{i,j} = \exp \left( \frac{\boldsymbol{\alpha}_i + \boldsymbol{\beta}_j - \tilde{\mathbf{C}}_{i,j}}{\epsilon} \right) \tag{13}$$

for an optimal transport map $\mathbf{T}$ to the entropy-regularized OT problem. We take the differential of Eq. (13) as follows

$$\mathrm{d}\tilde{\mathbf{C}}_{i,j} + \epsilon \mathrm{d} \left( \log \mathbf{T}_{i,j} \right) - \mathrm{d}\boldsymbol{\alpha}_i - \mathrm{d}\boldsymbol{\beta}_j = 0 \tag{14}$$

which can be reformulated in matrix form as follows

$$\mathrm{d}\mathbf{T} = \frac{1}{\epsilon} \mathbf{T} \odot \left( \mathrm{d}\boldsymbol{\alpha} \mathbf{1}_m^\top + \mathbf{1}_n \mathrm{d}\boldsymbol{\beta}^\top - \mathrm{d}\tilde{\mathbf{C}} \right) \tag{15}$$

Next, we impose the marginal constraints to derive a linear system of equations. Since the marginal constraints of OT are always satisfied, their differentials must be zero, i.e.,

$$\mathrm{d}\mathbf{T} \mathbf{1}_m = 0 \tag{16}$$

$$\left( \mathrm{d}\mathbf{T} \right)^\top \mathbf{1}_n = 0 \tag{17}$$

Firstly, we plug Eq. (15) into Eq. (16), which gives

$$\frac{1}{\epsilon} \left( \left( \mathbf{T} \odot \left( \mathrm{d}\boldsymbol{\alpha} \mathbf{1}_m^\top \right) \right) \mathbf{1}_m + \left( \mathbf{T} \odot \left( \mathbf{1}_n \mathrm{d}\boldsymbol{\beta}^\top \right) \right) \mathbf{1}_m - \left( \mathbf{T} \odot \mathrm{d}\tilde{\mathbf{C}} \right) \mathbf{1}_m \right) = 0 \tag{18}$$

where

$$\begin{cases} \left( \mathbf{T} \odot \left( \mathrm{d}\boldsymbol{\alpha} \mathbf{1}_m^\top \right) \right) \mathbf{1}_m = \left( \mathbf{T} \mathbf{1}_m \right) \odot \mathrm{d}\boldsymbol{\alpha} = diag(\boldsymbol{\mu}) \mathrm{d}\boldsymbol{\alpha} \\ \left( \mathbf{T} \odot \left( \mathbf{1}_n \mathrm{d}\boldsymbol{\beta}^\top \right) \right) \mathbf{1}_m = \mathbf{T} \mathrm{d}\boldsymbol{\beta} \end{cases} \tag{19}$$

Therefore, Eq (18) gives the first part of the linear system as follows

$$diag(\boldsymbol{\mu}) \mathrm{d}\boldsymbol{\alpha} + \mathbf{T} \mathrm{d}\boldsymbol{\beta} = \left( \mathbf{T} \odot \mathrm{d}\tilde{\mathbf{C}} \right) \mathbf{1}_m \tag{20}$$

Secondly, we plug Eq. (15) into Eq. (17), which gives

$$\frac{1}{\epsilon}\left(\left(\mathbf{T}\odot\left(d\boldsymbol{\alpha}\mathbf{1}_m^\top\right)\right)^\top\mathbf{1}_n + \left(\mathbf{T}\odot\left(\mathbf{1}_n d\boldsymbol{\beta}^\top\right)\right)^\top\mathbf{1}_n - \left(\mathbf{T}\odot d\tilde{\mathbf{C}}\right)^\top\mathbf{1}_n\right) = 0 \tag{21}$$

where

$$\begin{cases} \left(\mathbf{T}\odot\left(d\boldsymbol{\alpha}\mathbf{1}_m^\top\right)\right)^\top\mathbf{1}_n = \mathbf{T}^\top d\boldsymbol{\alpha} \\ \left(\mathbf{T}\odot\left(\mathbf{1}_n d\boldsymbol{\beta}^\top\right)\right)^\top\mathbf{1}_n = \left(\mathbf{T}^\top\mathbf{1}_n\right)\odot d\boldsymbol{\beta} = \mathrm{diag}(\boldsymbol{\nu})d\boldsymbol{\beta} \end{cases} \tag{22}$$

Therefore, Eq (21) gives the second part of the linear system as follows

$$\mathbf{T}^\top d\boldsymbol{\alpha} + \mathrm{diag}(\boldsymbol{\nu})d\boldsymbol{\beta} = \left(\mathbf{T}\odot d\tilde{\mathbf{C}}\right)^\top\mathbf{1}_n \tag{23}$$

Combining Eq. (20) and (23) gives a block linear system as follows

$$\begin{bmatrix}\mathrm{diag}(\boldsymbol{\mu}) & \mathbf{T} \\ \mathbf{T}^\top & \mathrm{diag}(\boldsymbol{\nu})\end{bmatrix}\begin{bmatrix}d\boldsymbol{\alpha} \\ d\boldsymbol{\beta}\end{bmatrix} = \begin{bmatrix}\left(\mathbf{T}\odot d\tilde{\mathbf{C}}\right)\mathbf{1}_m \\ \left(\mathbf{T}\odot d\tilde{\mathbf{C}}\right)^\top\mathbf{1}_n\end{bmatrix} \tag{24}$$

Now, we derive the differential of utility function $f(\mathbf{T}) : \mathbb{R}^{n\times m}\to\mathbb{R}$. By definition, $df = \langle\nabla_{\mathbf{T}}f, d\mathbf{T}\rangle$. We plug this equation into Eq. (15), which gives

$$df = \frac{1}{\epsilon}\left(\underbrace{\langle\mathbf{G}\mathbf{1}_m, d\boldsymbol{\alpha}\rangle + \langle\mathbf{G}^\top\mathbf{1}_n, d\boldsymbol{\beta}\rangle}_{\Omega} - \langle\mathbf{G}, d\tilde{\mathbf{C}}\rangle\right) \tag{25}$$

where $\mathbf{G}$ is defined as

$$\mathbf{G} := \mathbf{T}\odot\nabla_{\mathbf{T}}f \tag{26}$$

At this point, both $d\boldsymbol{\alpha}$ and $d\boldsymbol{\beta}$ are implicitly defined by Eq. (24). To eliminate $d\boldsymbol{\alpha}, d\boldsymbol{\beta}$, we introduce an adjoint vector $\mathbf{y}\in\mathbb{R}^{n+m}$ and define the following linear system in Eq. (27). For readability, We write $\mathbf{y} = \begin{bmatrix}\mathbf{y}_\alpha \\ \mathbf{y}_\beta\end{bmatrix}$ where $\mathbf{y}_\alpha\in\mathbb{R}^n, \mathbf{y}_\beta\in\mathbb{R}^m$.

$$\begin{bmatrix}\mathrm{diag}(\boldsymbol{\mu}) & \mathbf{T} \\ \mathbf{T}^\top & \mathrm{diag}(\boldsymbol{\nu})\end{bmatrix}\begin{bmatrix}\mathbf{y}_\alpha \\ \mathbf{y}_\beta\end{bmatrix} = \begin{bmatrix}\mathbf{G}\mathbf{1}_m \\ \mathbf{G}^\top\mathbf{1}_n\end{bmatrix} \tag{27}$$

Let

$$\mathbf{A} = \begin{bmatrix}\mathrm{diag}(\boldsymbol{\mu}) & \mathbf{T} \\ \mathbf{T}^\top & \mathrm{diag}(\boldsymbol{\nu})\end{bmatrix} \tag{28}$$

For Eq. (24) and (27), we have

$$\mathbf{A}\begin{bmatrix}d\boldsymbol{\alpha} \\ d\boldsymbol{\beta}\end{bmatrix} = \begin{bmatrix}\left(\mathbf{T}\odot d\tilde{\mathbf{C}}\right)\mathbf{1}_m \\ \left(\mathbf{T}\odot d\tilde{\mathbf{C}}\right)^\top\mathbf{1}_n\end{bmatrix}, \quad \mathbf{A}\begin{bmatrix}\mathbf{y}_\alpha \\ \mathbf{y}_\beta\end{bmatrix} = \begin{bmatrix}\mathbf{G}\mathbf{1}_m \\ \mathbf{G}^\top\mathbf{1}_n\end{bmatrix} \tag{29}$$

Now, we can reformulate the $\Omega$ term in Eq (25) as follows

$$\begin{aligned} \Omega = \langle\mathbf{G}\mathbf{1}_m, d\boldsymbol{\alpha}\rangle + \langle\mathbf{G}^\top\mathbf{1}_n, d\boldsymbol{\beta}\rangle &= \left\langle\begin{bmatrix}\mathbf{G}\mathbf{1}_m \\ \mathbf{G}^\top\mathbf{1}_n\end{bmatrix}, \begin{bmatrix}d\boldsymbol{\alpha} \\ d\boldsymbol{\beta}\end{bmatrix}\right\rangle \\ &= \underbrace{\left\langle\mathbf{A}\begin{bmatrix}\mathbf{y}_\alpha \\ \mathbf{y}_\beta\end{bmatrix}, \begin{bmatrix}d\boldsymbol{\alpha} \\ d\boldsymbol{\beta}\end{bmatrix}\right\rangle = \left\langle\begin{bmatrix}\mathbf{y}_\alpha \\ \mathbf{y}_\beta\end{bmatrix}, \mathbf{A}\begin{bmatrix}d\boldsymbol{\alpha} \\ d\boldsymbol{\beta}\end{bmatrix}\right\rangle}_{\text{given that } \mathbf{A}=\mathbf{A}^\top} \\ &= \left\langle\begin{bmatrix}\mathbf{y}_\alpha \\ \mathbf{y}_\beta\end{bmatrix}, \begin{bmatrix}\left(\mathbf{T}\odot d\tilde{\mathbf{C}}\right)\mathbf{1}_m \\ \left(\mathbf{T}\odot d\tilde{\mathbf{C}}\right)^\top\mathbf{1}_n\end{bmatrix}\right\rangle = \langle\mathbf{y}_\alpha, \left(\mathbf{T}\odot d\tilde{\mathbf{C}}\right)\mathbf{1}_m\rangle + \langle\mathbf{y}_\beta, \left(\mathbf{T}\odot d\tilde{\mathbf{C}}\right)^\top\mathbf{1}_n\rangle \end{aligned} \tag{30}$$

Therefore, we reformulate Eq (25) as follows

$$\begin{aligned} \mathrm{d}f &= \frac{1}{\epsilon}\left(\langle \mathbf{y}_\alpha, \left(\mathbf{T}\odot \mathrm{d}\tilde{\mathbf{C}}\right)\mathbf{1}_m\rangle + \langle \mathbf{y}_\beta, \left(\mathbf{T}\odot \mathrm{d}\tilde{\mathbf{C}}\right)^\top \mathbf{1}_n\rangle - \langle \mathbf{G}, \mathrm{d}\tilde{\mathbf{C}}\rangle\right) \\ &= \frac{1}{\epsilon}\left(\langle \mathbf{T}\odot \left(\mathbf{y}_\alpha \mathbf{1}_m^\top\right), \mathrm{d}\tilde{\mathbf{C}}\rangle + \langle \mathbf{T}\odot \left(\mathbf{1}_n \mathbf{y}_\beta^\top\right), \mathrm{d}\tilde{\mathbf{C}}\rangle - \langle \mathbf{G}, \mathrm{d}\tilde{\mathbf{C}}\rangle\right) \\ &= \left\langle \frac{1}{\epsilon}\mathbf{T}\odot \left(\mathbf{y}_\alpha \mathbf{1}_m^\top + \mathbf{1}_n \mathbf{y}_\beta^\top - \nabla_\mathbf{T}f\right), \mathrm{d}\tilde{\mathbf{C}}\right\rangle \end{aligned} \tag{31}$$

In this way, we have the final formulation of $\nabla_{\tilde{\mathbf{C}}}f$ as follows

$$\nabla_{\tilde{\mathbf{C}}}f = \frac{\mathrm{d}f}{\mathrm{d}\tilde{\mathbf{C}}} = \frac{1}{\epsilon}\mathbf{T}\odot \left(\mathbf{y}_\alpha \mathbf{1}_m^\top + \mathbf{1}_n \mathbf{y}_\beta^\top - \nabla_\mathbf{T}f\right) \tag{32}$$

s.t.

$$\begin{bmatrix} \mathrm{diag}(\boldsymbol{\mu}) & \mathbf{T} \\ \mathbf{T}^\top & \mathrm{diag}(\boldsymbol{\nu}) \end{bmatrix}\begin{bmatrix} \mathbf{y}_\alpha \\ \mathbf{y}_\beta \end{bmatrix} = \begin{bmatrix} \mathbf{G}\mathbf{1}_m \\ \mathbf{G}^\top \mathbf{1}_n \end{bmatrix} = \begin{bmatrix} (\mathbf{T}\odot \nabla_\mathbf{T}f)\mathbf{1}_m \\ (\mathbf{T}\odot \nabla_\mathbf{T}f)^\top \mathbf{1}_n \end{bmatrix} \tag{33}$$

$\square$

## A.2. Proof of Singularity of A in Eq. (6)

*Proof.* By Eq. (6), we have

$$\mathbf{A} = \begin{bmatrix} \mathrm{diag}(\boldsymbol{\mu}) & \mathbf{T} \\ \mathbf{T}^\top & \mathrm{diag}(\boldsymbol{\nu}) \end{bmatrix} \tag{34}$$

where $\boldsymbol{\mu}, \boldsymbol{\nu}$ are marginal distributions of OT, and $\mathbf{T}$ is the solved transport map. As $\mathbf{T}$ satisfy the marginal constraints of OT, i.e., $\mathbf{T}\mathbf{1}_m = \boldsymbol{\mu}$, $\mathbf{T}^\top \mathbf{1}_n = \boldsymbol{\nu}$, we can find a nonzero vector $\begin{bmatrix} \mathbf{1}_n \\ -\mathbf{1}_m \end{bmatrix}$ in the null space of $\mathbf{A}$, i.e.,

$$\mathbf{A}\mathbf{z} = \begin{bmatrix} \mathrm{diag}(\boldsymbol{\mu}) & \mathbf{T} \\ \mathbf{T}^\top & \mathrm{diag}(\boldsymbol{\nu}) \end{bmatrix}\begin{bmatrix} \mathbf{1}_n \\ -\mathbf{1}_m \end{bmatrix} = \begin{bmatrix} \mathrm{diag}(\boldsymbol{\mu})\mathbf{1}_n - \mathbf{T}\mathbf{1}_m \\ \mathbf{T}^\top \mathbf{1}_n - \mathrm{diag}(\boldsymbol{\nu})\mathbf{1}_m \end{bmatrix} = \begin{bmatrix} \boldsymbol{\mu} - \mathbf{T}\mathbf{1}_m \\ \mathbf{T}^\top \mathbf{1}_n - \boldsymbol{\nu} \end{bmatrix} = \mathbf{0}_{n+m} \tag{35}$$

Therefore, $\mathbf{A}$ is a singular matrix.

$\square$

## A.3. Proof of Lemma 4.3

**Lemma.** *Eq. (6) can be solved via conjugate gradient method with guaranteed convergence to global optimum.*

*Proof.* First, we prove that Eq. (6) can be solved via conjugate gradient method with guaranteed convergence. While the classical CG method requires the coefficient matrix $\mathbf{A}$ to be strictly positive definite, (Kaasschieter, 1988; Hayami, 2018) show that CG method can be applied directly to singular systems with guaranteed convergence when (1) $\mathbf{A}$ is positive-semidefinite, and (2) vector $\mathbf{b}$ in Eq. (6) lies in the range $R(\mathbf{A})$ of $\mathbf{A}$.

We begin by proving that matrix $\mathbf{A}$ in Eq. (6) is positive-semidefinite. Firstly, $\mathbf{A} \in \mathbb{R}^{(n+m)\times(n+m)}$ is symmetric as $\mathbf{A} = \mathbf{A}^\top$. Secondly, for any vectors $\mathbf{s} \in \mathbb{R}^n, \mathbf{t} \in \mathbb{R}^m$, we have

$$\begin{bmatrix} \mathbf{s} \\ \mathbf{t} \end{bmatrix}^\top \mathbf{A}\begin{bmatrix} \mathbf{s} \\ \mathbf{t} \end{bmatrix} = \begin{bmatrix} \mathbf{s} \\ \mathbf{t} \end{bmatrix}^\top \begin{bmatrix} \mathrm{diag}(\boldsymbol{\mu}) & \mathbf{T} \\ \mathbf{T}^\top & \mathrm{diag}(\boldsymbol{\nu}) \end{bmatrix}\begin{bmatrix} \mathbf{s} \\ \mathbf{t} \end{bmatrix} = \mathbf{s}^\top \mathrm{diag}(\boldsymbol{\mu})\mathbf{s} + 2\mathbf{s}^\top \mathbf{T}\mathbf{t} + \mathbf{t}^\top \mathrm{diag}(\boldsymbol{\nu})\mathbf{t} \tag{36}$$

As $\mathbf{T}$ satisfy the marginal constraints of OT, i.e., $\sum_{j=1}^m \mathbf{T}_{i,j} = \boldsymbol{\mu}_i$, $\sum_{i=1}^n \mathbf{T}_{i,j} = \boldsymbol{\nu}_j$, we have

$$\mathbf{s}^\top \mathrm{diag}(\boldsymbol{\mu})\mathbf{s} = \sum_{i=1}^n \mathbf{s}_i^2 \boldsymbol{\mu}_i = \sum_{i=1}^n \mathbf{s}_i^2 \sum_{j=1}^m \mathbf{T}_{i,j} \tag{37}$$

$$2\mathbf{s}^\top \mathbf{T}\mathbf{t} = 2\sum_{i=1}^n \sum_{j=1}^m \mathbf{s}_i \mathbf{t}_j \mathbf{T}_{i,j} \tag{38}$$

$$\mathbf{t}^\top \mathrm{diag}(\boldsymbol{\nu})\mathbf{t} = \sum_{j=1}^m \mathbf{t}_j^2 \boldsymbol{\nu}_j = \sum_{j=1}^m \mathbf{t}_j^2 \sum_{i=1}^n \mathbf{T}_{i,j} \tag{39}$$

Therefore, Eq. (36) can be reformulated to

$$\begin{bmatrix} \mathbf{s} \\ \mathbf{t} \end{bmatrix}^\top \mathbf{A} \begin{bmatrix} \mathbf{s} \\ \mathbf{t} \end{bmatrix} = \sum_{i=1}^{n} \sum_{j=1}^{m} \mathbf{T}_{i,j} \left( \mathbf{s}_i^2 + 2\mathbf{s}_i\mathbf{t}_j + \mathbf{t}_j^2 \right) = \sum_{i=1}^{n} \sum_{j=1}^{m} \mathbf{T}_{i,j} \left( \mathbf{s}_i + \mathbf{t}_j \right)^2 \geq 0 \tag{40}$$

In this case, we have proven that $\mathbf{A}$ in Eq. (6) is positive-semidefinite.

Then, we prove that vector $\mathbf{b}$ in Eq. (6) lies in the range $R(\mathbf{A})$ of $\mathbf{A}$ by showing that $\mathbf{b}$ is orthogonal to the null space of $\mathbf{A}$, i.e., $\mathbf{z}^\top \mathbf{b} = 0$, $\forall \mathbf{z} \in \text{Null}(\mathbf{A})$ (Strang, 2022). For a vector $\begin{bmatrix} \mathbf{s} \in \mathbb{R}^n \\ \mathbf{t} \in \mathbb{R}^m \end{bmatrix}$ in the null space of $\mathbf{A}$, we have $\mathbf{A} \begin{bmatrix} \mathbf{s} \\ \mathbf{t} \end{bmatrix} = \mathbf{0}_{n+m}$, which means that $\begin{bmatrix} \mathbf{s} \\ \mathbf{t} \end{bmatrix}^\top \mathbf{A} \begin{bmatrix} \mathbf{s} \\ \mathbf{t} \end{bmatrix} = 0$. As we shown in Appendix A.3, for any vector $\begin{bmatrix} \mathbf{s} \\ \mathbf{t} \end{bmatrix}$, we have $\begin{bmatrix} \mathbf{s} \\ \mathbf{t} \end{bmatrix}^\top \mathbf{A} \begin{bmatrix} \mathbf{s} \\ \mathbf{t} \end{bmatrix} = \sum_{i=1}^{n} \sum_{j=1}^{m} \mathbf{T}_{i,j} \left( \mathbf{s}_i + \mathbf{t}_j \right)^2 \geq 0$, therefore the null space of $\mathbf{A}$ is

$$\text{Null}(\mathbf{A}) = \left\{ \begin{bmatrix} \mathbf{s} \\ \mathbf{t} \end{bmatrix} : \mathbf{s}_i + \mathbf{t}_j = 0 \right\} = \text{span} \left\{ \begin{bmatrix} \mathbf{1}_n \\ -\mathbf{1}_m \end{bmatrix} \right\} \tag{41}$$

Since

$$\begin{bmatrix} \mathbf{1}_n \\ -\mathbf{1}_m \end{bmatrix}^\top \mathbf{b} = \begin{bmatrix} \mathbf{1}_n \\ -\mathbf{1}_m \end{bmatrix}^\top \begin{bmatrix} (\mathbf{T} \odot \nabla_{\mathbf{T}} f) \mathbf{1}_m \\ (\mathbf{T} \odot \nabla_{\mathbf{T}} f)^\top \mathbf{1}_n \end{bmatrix} = 0 \tag{42}$$

then $\mathbf{b}$ lies in the range $R(\mathbf{A})$ of $\mathbf{A}$. In this way, we have proven that CG solves Eq. (6) with guaranteed convergence.

Then, we prove that CG converges to the global optimum of Eq. (6). As $\mathbf{A}$ is symmetric, solving the linear system $\mathbf{Ay} = \mathbf{b}$ is equivalent to minimizing a quadratic function $g$ as follows (Boyd & Vandenberghe, 2004)

$$g(\mathbf{y}) = \frac{1}{2} \mathbf{y}^\top \mathbf{A} \mathbf{y} - \mathbf{b}^\top \mathbf{y} \tag{43}$$

We can see that the Hessian of $g$ is $\mathbf{A}$, which is positive-semidefinite as proven above. Therefore, $g$ is a convex quadratic function w.r.t. $\mathbf{y}$ (Boyd & Vandenberghe, 2004). Since Eq. (6) corresponds to the first-order optimality condition $\nabla_{\mathbf{y}} g(\mathbf{y}) = 0$, any solution of the linear system is a global optimum (Boyd & Vandenberghe, 2004). Since the CG method converges to a solution of Eq. (6) as proven above, the solution must be the global optimum. □

### A.4. Proof of Theorem 4.5

**Theorem.** *(Time & Space Complexity of* AVATAR-$L_2$/ENTROPY/CONSIST*)*

*The time complexity is $\mathcal{O}\left( \frac{k}{b} K(n+m) \right)$ for* AVATAR-$L_2$/ENTROPY*, and $\mathcal{O}\left( \frac{k}{b}(K(n+m)+e) \right)$ for* AVATAR-CONSIST*, where $e$ denotes the number of edges in the networks to be aligned[2]. The space complexity of* AVATAR-$L_2$/ENTROPY/ CONSIST *is $\mathcal{O}(nm)$. $k$ is the total query budget, $b$ is the batch query size, $K$ is the number of iterations of CG, and $n, m$ are number of objects in the source and target sets, respectively.*

*Proof.* The computation of AVATAR consist of two main steps: 1) compute the pairwise object query impact by Eq. (7) 2) compute the posterior object query impact by Eq. (8). Note that the transport map $\mathbf{T}$ at the end of each query round is typically a deterministic and *sparse* matrix with $\mathcal{O}(n+m)$ non-zero entries for alignment problems (Chen et al., 2020; Zeng et al., 2023a; Yu et al., 2025a), as shown empirically in Figure 11, making $\mathbf{A}$ in Eq. (6) sparse as well. We denote the number of non-zero entries in $\mathbf{T}$ by nnz$(\mathbf{T})$. For the first step, we need to compute the vector $\mathbf{b}$ in Eq. (6) first in $\mathcal{O}(\text{nnz}(\mathbf{T}))$ for AVATAR-$L_2$/ENTROPY, and $\mathcal{O}(\text{nnz}(\mathbf{T}) + n + m + e)$ for AVATAR-CONSIST (Macskassy, 2009). Then, we solve the linear system in Eq. (6) via the CG method with a time complexity of $\mathcal{O}(K\text{nnz}(\mathbf{T}))$, where $K$ is the total number of CG iterations (Kaasschieter, 1988). For the second step, Eq. (8) aggregates pairwise query impact in $\mathcal{O}(\text{nnz}(\mathbf{T}))$. Therefore, for AVATAR-$L_2$/ENTROPY, the time complexity for one query batch is $\mathcal{O}(\text{nnz}(\mathbf{T})) + \mathcal{O}(K\text{nnz}(\mathbf{T})) + \mathcal{O}(\text{nnz}(\mathbf{T})) = \mathcal{O}(K\text{nnz}(\mathbf{T})) \approx \mathcal{O}(K(n+m))$, making the total time complexity $\mathcal{O}(\frac{k}{b}K(n+m))$; for AVATAR-CONSIST, the time complexity for one query batch is $\mathcal{O}(\text{nnz}(\mathbf{T}) + n + m + e) + \mathcal{O}(K\text{nnz}(\mathbf{T})) + \mathcal{O}(\text{nnz}(\mathbf{T})) = \mathcal{O}(K\text{nnz}(\mathbf{T}) + n + m + e) \approx \mathcal{O}(K(n+m)+e)$, making the total time complexity $\mathcal{O}\left( \frac{k}{b}(K(n+m)+e) \right)$

While the transport map $\mathbf{T}$ is typically sparse for AVATAR, Eq. (7) requires explicit storage of the dense transport cost matrix $\mathbf{C}$. Therefore, the space complexity of AVATAR-$L_2$ is $\mathcal{O}(nm)$. □

---

[2]Without loss of generality, we assume $\mathcal{O}(e) \approx \mathcal{O}(e_1) \approx \mathcal{O}(e_2)$ where $e_i$ denotes the number of edges in the $i$-th networks (Zeng et al., 2023a; Yu et al., 2025b).

## A.5. Proof of Theorem 4.6

**Theorem A.1.** *(Convergence of AVATAR)*

*The conjugate gradient method applied to the linear system of Eq. (6) converges at a linear convergence rate of $\frac{\sqrt{\lambda_1/\lambda_r}-1}{\sqrt{\lambda_1/\lambda_r}+1}$, where $\lambda_1, \lambda_r$ denotes the largest/smallest nonzero eigenvalues of $\mathbf{A}$ in Eq. (6).*

*Proof.* (Hayami, 2018) shows that the error bound of CG method on a singular system $\mathbf{A}\mathbf{y} = \mathbf{b}$ is

$$\|\mathbf{y}^{(k)} - \mathbf{y}^*\|_{\mathbf{\Lambda}_r} \leq 2 \left\{ \frac{\sqrt{\kappa(\mathbf{\Lambda}_r)} - 1}{\sqrt{\kappa(\mathbf{\Lambda}_r)} + 1} \right\}^k \|\mathbf{y}^{(0)} - \mathbf{y}^*\|_{\mathbf{\Lambda}_r} \tag{44}$$

where $\mathbf{y}^{(k)}$ denotes $\mathbf{y}$ in the $k$-th round of CG method, and $\mathbf{y}^*$ denotes the optimal solution. $\|\mathbf{x}\|_{\mathbf{\Lambda}_r} = \mathbf{x}^\top \mathbf{\Lambda}_r \mathbf{x}$, where $\mathbf{\Lambda}_r$ is a diagonal matrix of the nonzero eigenvalues of $\mathbf{A}$, i.e.,

$$\mathbf{\Lambda}_r = \begin{bmatrix} \lambda_1 & & \\ & \ddots & \\ & & \lambda_r \end{bmatrix}, \ \lambda_1 \geq \lambda_2 \geq \ldots \geq \lambda_r > 0 \tag{45}$$

where $r = \text{rank}(\mathbf{A})$, $\lambda_i$ are the nonzero eigenvalues of $\mathbf{A}$. $\kappa(\mathbf{\Lambda}_r) = \frac{\lambda_1}{\lambda_r} \geq 1$. Since $\rho = \frac{\sqrt{\kappa(\mathbf{\Lambda}_r)}-1}{\sqrt{\kappa(\mathbf{\Lambda}_r)}+1} \in [0,1)$ in this case, Eq. (6) converges linearly to a solution by CG method (Kaasschieter, 1988). □

# B. Detailed Experimental Pipeline

## B.1. Datasets Descriptions

Detailed descriptions of datasets adopted in this paper are given as follows.

**Phone-Email (Zhang et al., 2017).** A pair of communication networks with nodes representing people and edges representing documented communication between them via phone or email. The phone network contains 1,000 nodes and 41,191 edges, and the Email network contains 1,003 nodes and 4,627 edges. No attribute information is available for both networks. There are 1,000 common people across two networks as ground-truth alignment.

**ACM-DBLP-P(A) (Tang et al., 2008).** A pair of undirected co-authorship networks with nodes representing authors and edges representing co-authorship between two authors. The ACM network contains 9,872 nodes and 39,561 edges, and the DBLP network contains 9,916 nodes and 44,808 edges. ACM-DBLP-A contains node attribute information while ACM-DBLP-P are plain networks. There are 6,325 common authors across two networks as ground-truth alignment.

**Douban (Zhang & Tong, 2016).** A pair of online-offline social networks collected from Douban, with nodes representing users and edges representing user interactions on the website. The online network of Douban contains 3,906 nodes and 8,164 edges, and the offline network of Douban contains 1,118 nodes and 1,511 edges. Node attribute are constructed from the the location of a user, and edge attributes from the contact/friend relationship on the social platform. There are 1,118 common user across the two networks.

**CIFAR-10-C (Hendrycks & Dietterich, 2019).** CIFAR-10-C consist of the original CIFAR-10 (Krizhevsky et al., 2009) test set images transformed by different types of corruptions at five levels of severity. In this paper, we adopt the gaussian perturbation version of CIFAR-10-C with level 5 severity. For image-text retrieval on CIFAR-10-C, we construct image-text pairs by associating each images with the corresponding textual description based on its class label, and treat retrieval as matching images to their corresponding text embeddings.

**ImageNet-C (Hendrycks & Dietterich, 2019).** ImageNet-C consist of the original images from ImageNet transformed by different types of corruptions at five levels of severity. we adopt the gaussian perturbation version level 5 severity. Image-text retrieval on ImageNet-C follows the same paradigm as CIFAR10-C.

**COCO (Lin et al., 2014).** COCO is a large-scale dataset for object detection, segmentation, and captioning. It contains photos of 91 objects types and around 2,500,000 labeled instances in 328,000 images. For image-text grounding on COCO, which contains explicit fine-grained correspondence between phrases in a sentence and objects (or regions) in an image, we treat grounding as an matching problem between phrases and objects for an image.

**Flickr30K Entities (Plummer et al., 2015).** Flickr30K Entities is a standard benchmark for sentence-based image description, constructed by augmenting around 158,000 captions from the Flickr30k (Young et al., 2014) dataset. Flickr30K Entities links the same entities across different captions for the same image, and associating them with manually annotated bounding boxes in Flickr30K. Image-text grounding on Flickr30K Entities follows the same paradigm as COCO.

### B.2. Introduction of Baseline Query Strategies

**Random.** Selects candidate object from $\mathcal{X}$ randomly.

**Entropy (Ren et al., 2021).** Select candidate object from $\mathcal{X}$ whose Shannon entropy of its alignment results, i.e., $\sum_j^m -\mathbf{T}_{i,j} \log \mathbf{T}_{i,j}$ for $x_i \in \mathcal{X}$, is the largest. The intuition is to select the most uncertain object to query, measured by entropy of its alignment results.

**Margin (Ren et al., 2021).** Select candidate objects from $\mathcal{X}$ whose differences between the two most probable labels reflected by $\mathbf{T}$, i.e., $\mathbf{T}_{i,j_1} - \mathbf{T}_{i,j_2}$ where $\mathbf{T}_{i,j_1} := \arg\max_j \mathbf{T}_{i,j}$ and $\mathbf{T}_{i,j_2} := \arg\max_{j \neq j_1} \mathbf{T}_{i,j}$, is the smallest. The intuition is to select the most uncertain object to query, measured by such differences.

**Least Confident (Zhou et al., 2021).** Select candidate objects from $\mathcal{X}$ whose confidence of alignment measured by the corresponding probability in $\mathbf{T}$, i.e., $\max_j \mathbf{T}_{i,j}$, is the smallest. The intuition is to select the least confident object to query.

**Betweenness (Zhou et al., 2021).** Select candidate nodes of NA from $\mathcal{X}$ that has the largest betweenness centrality scores. (Freeman, 1977).

**Density (Li et al., 2024).** Select candidate nodes from $\mathcal{X}$ that can represent all unlabeled source objects. The density score of a candidate object $x$ is defined as $\text{Density}(x_i) = \sum_{x_j \in \mathcal{U} \cup \mathcal{N}(x_i, k)}^{k} \|\mathbf{T}_{i,:} - \mathbf{T}_{j,:}\|_2^2$, where $\mathcal{U}$ denotes the set of unlabeled source objects, and $\mathcal{N}(x_i, k)$ denotes the $k$ nearest neighbors of $x_i$ in $\mathcal{X}$. We adopt $k = 20$ (Kim & Shin, 2022). Candidate objects with the highest density score are selected for query.

**Diversity (Li et al., 2024).** Select candidate nodes from $\mathcal{X}$ that are different from labeled source objects. The density score of a candidate object $x$ is defined as $\text{Density}(x_i) = \sum_{x_j \in \mathcal{L}} \text{KL}(\mathbf{T}_{i,:}, \mathbf{T}_{j,:})$, where $\mathcal{L} \subset \mathcal{X}$ denotes the set of labeled source objects, and $\text{KL}(\cdot, \cdot)$ denotes the KL divergence.

**GibbsMatchings & TopMatchings (Malmi et al., 2017).** Two matching-based query strategies for active NA that select uncertain candidate nodes from $\mathcal{X}$ whose sampled matchings aligns to different target nodes during different samples.

### B.3. Detailed Experimental Setup

**Metrics.** We adopt MRR and Recall@1 as the benchmarking metrics. For **mean reciprocal rank (MRR)**, it is defined as the average reciprocal of the rank at which the correct alignment appears in the candidate list, i.e., MRR$= \frac{1}{n} \sum_{i=1}^n \frac{1}{\text{rank}_i}$, where $n$ is the size of source object set, and rank$_i$ is the rank of the correct alignment for the $i$-th object in the source set; for **Recall@1**, it is defined as the proportion of source object whose correct alignment is ranked 1st by an alignment method, i.e., Recall@1$= \frac{1}{n} \sum_{i=1}^n \mathbb{I}\{\text{rank}_i = 1\}$, where $\mathbb{I}\{\cdot\}$ denotes the indicator function. For the **total query time**, it is defined as the accumulated runtime of different active alignment methods under a fixed query budget $k$ and query batch size $n_b$. Formally, $t^{(k, n_b)} = \sum_{i=1}^N t_i^{(k, n_b)}$, where $t_i^{(k, n_b)}$ is time required to select query candidates at the i-th round, after the completion of the alignment algorithm. $N$ denotes the total number of query rounds.

**Reproducibility.** For all experiments, the reported results are averaged against 5 different runs which randomly selects 20% ground-truth alignment as prior supervision for each run. To ensure a fair comparison, all baselines are given the same total query budget of 20% ground-truth, averaged against 10 query rounds (batches). Hyperparameters of all baselines alignment methods are set as default in their official implementations. Code and datasets are available at https://github.com/yq-leo/AvAtar-ICML26.

**Machine.** All experiments are conducted on a server with dual Intel® Xeon® Gold 6240R CPUs and 4 NVIDIA Tesla V100-SXM2 GPUs of 32GB memory.

# C. Additional Experimental Results

## C.1. Detailed Effectiveness Results

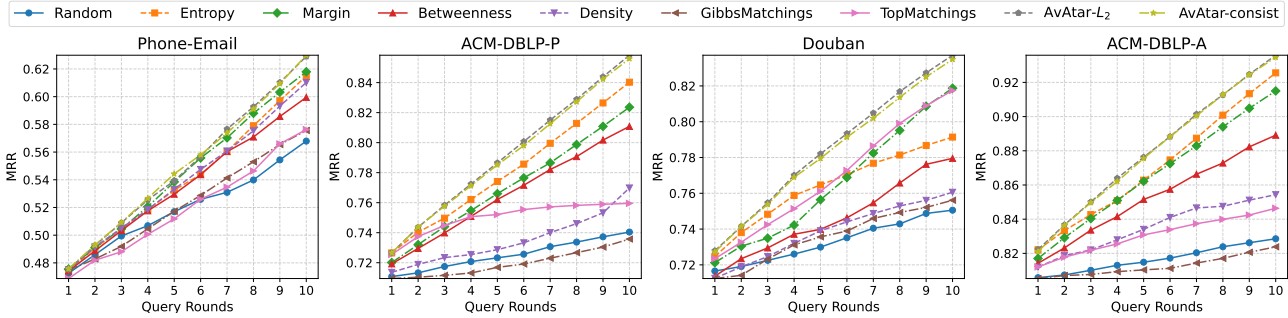

*Figure 6.* MRR vs. query round on four NA datasets using the PARROT (Zeng et al., 2023a) algorithm

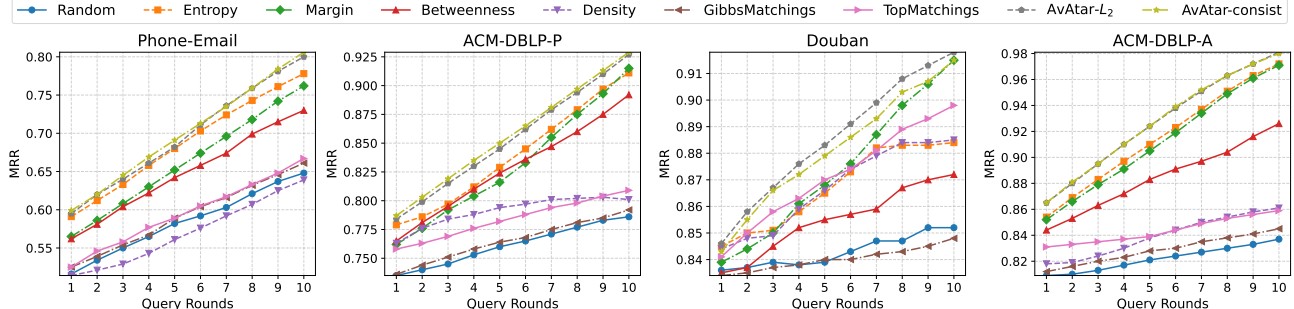

*Figure 7.* MRR vs. query round on four NA datasets using the JOENA (Yu et al., 2025b) algorithm

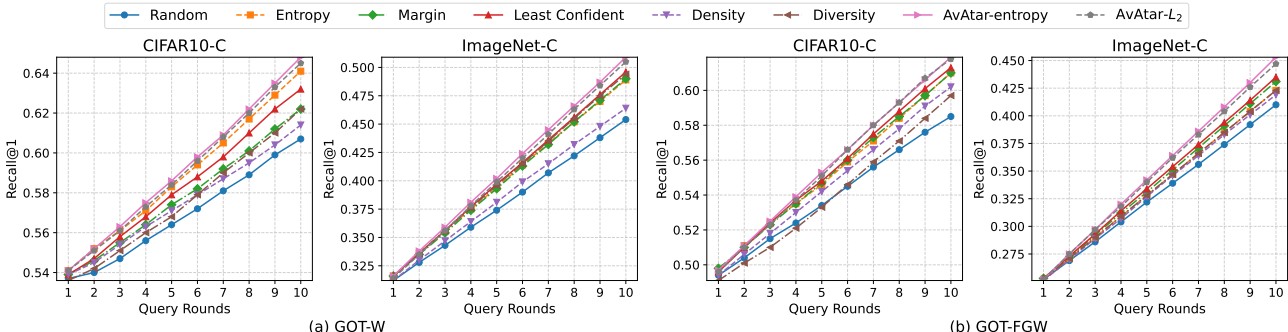

*Figure 8.* MRR vs. query round on two image-text retrieval datasets using the GOT (Chen et al., 2020) algorithm

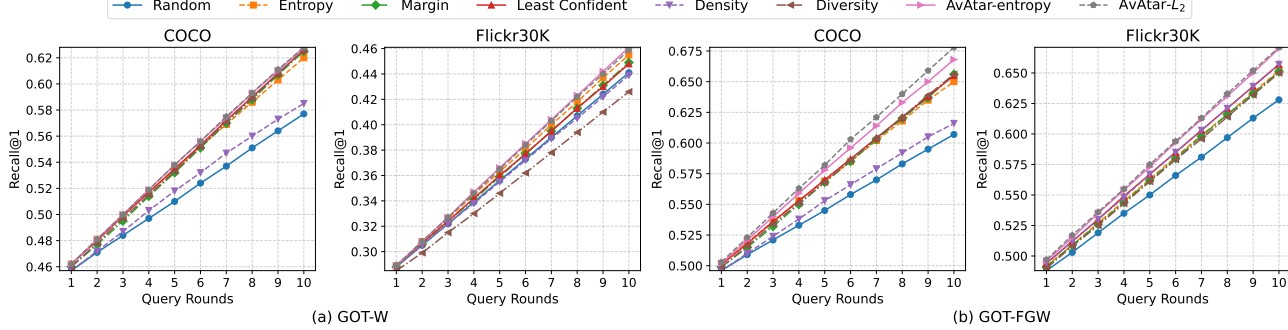

*Figure 9.* MRR vs. query round on two image-text grounding datasets using the GOT (Chen et al., 2020) algorithm

## C.2. Scalability Results

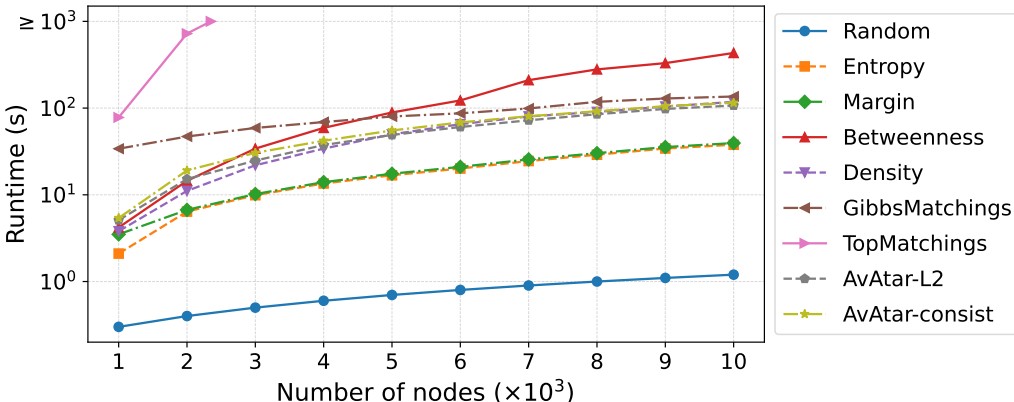

*Figure 10.* **Scalability results (under $10^3$ runtime limit) on synthetic graphs generated by the Erdős–Rényi (ER) model with an average node degree of 10.** The **x-axis** shows the number of nodes in the ER graphs (in $10^3$), and the **y-axis** of shows the runtime of query methods. The results confirm confirms that both AVATAR-$L_2$ and AVATAR-consist scales linearly w.r.t. the number of nodes in networks, making it scalable to large-scale alignment problems.

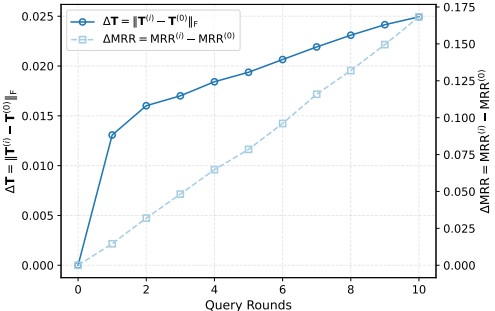

*Figure 11.* Drift of transport plan $\mathbf{T}$ and alignment performance improvement across query rounds, on Phone-Email dataset using the PARROT (Zeng et al., 2023a) algorithm and AVATAR-$L_2$

*Table 8.* Comparison of OT-based alignment algorithm (PARROT) + OT-specific active learning methods (AVATAR-$L_2$) with non-OT aligment algorithm (FINAL) + non-OT active learning methods (Attent) on NA tasks, in MRR.

| Dataset | Phone-Email | | Douban | |
|---|---|---|---|---|
| #-th Query Round | 5 | 10 | 5 | 10 |
| PARROT + AVATAR-$L_2$ | 0.539 | 0.629 | 0.782 | 0.837 |
| FINAL + Attent | 0.158 | 0.219 | 0.315 | 0.348 |
| $\Delta$ | +0.381 | +0.410 | +0.467 | +0.489 |

## C.3. Comparision with Non-OT Baselines

To further demonstrate the power of OT-based alignment algorithm, e.g., PARROT (Zeng et al., 2023a), equipped with OT-specific active learning approach, i.e., AVATAR, we compare its performance with a consistency-based alignment algorithm FINAL (Zhang & Tong, 2016) with an active learning approach Attent (Zhou et al., 2021) tailored for consistency-based methods. The results are shown in Table 8, which shows that **(1) PARROT + AVATAR consistently outperforms FINAL + Attent**, demonstrating the power of OT-based alignment. **(2) The performance gap increases as the query round (budget) increases**, showing the superiority of AVATAR in boosting the performance of OT-based alignment methods.

## C.4. Study on External Hyperparameters

We further validates the robustness of AVATAR against external hyperparameters of specific OT-based alignment algorithms, i.e., the penalizing factor $\beta$ and the entropic regularization parameter $\epsilon$, we conduct an additional set of hyperparameter study on Phone-Email with PARROT (Zeng et al., 2023a), and report results in Table 9.

# D. Limitations & Future Works

In this section, we discuss some of the potential limitations of the proposed AVATAR. Firstly, the benefit of AvAtar may be partially reduced in some geometric matching tasks, e.g., point cloud registration (PCR), where the cost functions are relatively robust due to rich geometric information. However, this does not not render AvAtar inapplicable to PCR,

*Table 9.* Study on external hyperparameters $\beta$ and $\epsilon$ on Phone-Email with PARROT. The best results are highlighted in **bold**.

| Hyperparameter | penalizing factor $\beta$ | | | | | | entropic regularization weight $\epsilon$ | | | | | |
|---|---|---|---|---|---|---|---|---|---|---|---|---|
| Values | 1.0 | | 0.5 | | 0.1 | | 6e-5 | | 6e-4 | | 6e-3 | |
| #-th Query Round | 5 | 10 | 5 | 10 | 5 | 10 | 5 | 10 | 5 | 10 | 5 | 10 |
| RANDOM | 0.517 | 0.568 | 0.290 | 0.335 | 0.289 | 0.321 | 0.352 | 0.394 | 0.517 | 0.568 | 0.218 | 0.252 |
| ENTROPY | 0.533 | 0.615 | 0.312 | 0.367 | 0.299 | 0.349 | 0.372 | 0.414 | 0.533 | 0.615 | 0.228 | 0.259 |
| MARGIN | 0.538 | 0.618 | 0.315 | 0.359 | 0.310 | 0.367 | 0.376 | 0.428 | 0.538 | 0.618 | 0.221 | 0.261 |
| TOPMATCHINGS | 0.512 | 0.576 | 0.308 | 0.329 | 0.301 | 0.352 | 0.358 | 0.409 | 0.512 | 0.576 | 0.231 | 0.279 |
| AVATAR-$L_2$ | **0.539** | **0.629** | **0.329** | **0.391** | **0.321** | **0.369** | **0.389** | **0.437** | **0.539** | **0.629** | **0.249** | **0.283** |

especially under imperfect cost design due to outlier points and noisy features (Qin et al., 2023; Qiu et al., 2024). Secondly, while AVATAR can be easily extended to some OT variants, e.g., unbalanced OT (Gabriel & Marco, 2019), and Gromov-Wasserstein distance (Peyré et al., 2016), with entropic regularization, its extension to neural OT (Korotin et al., 2022) remains an interesting future work.

