# OpenReview forum: "AvAtar: Learning to Align via Active Optimal Transport"
_ICML.cc/2026/Conference — ICML 2026 regular_

### Official Review · Reviewer_PnUW · 2026-03-01

**Soundness:** 2
**Presentation:** 3
**Significance:** 3
**Originality:** 3
**Overall Recommendation:** 4
**Confidence:** 4

**Summary:**

This paper investigates the active alignment problem within OT and introduces AVATAR, a novel method that identifies the most informative candidates for supervision. It quantifies candidate informativeness by measuring their impact on global alignment through gradient propagation across an entropy-regularized OT formulation. To address the computational challenges of differentiating with respect to OT constraints, the authors employ the adjoint-state method to reformulate the gradient computation as a linear system. Experiments across three diverse alignment tasks validate the effectiveness of the proposed AVATAR.

**Compliance With Llm Reviewing Policy:**

Affirmed.

**Final Justification:**

The authors’ explanations and additional experiments have addressed my concerns about the method's practicality. Therefore, I raise my score.

**Key Questions For Authors:**

1. Does the matrix $H$ serve as a discrete counterpart to the transport plan $T$, essentially acting as a supervisory signal for the optimization of $T$? Regarding computational efficiency: would it be more effective to prune or discard the data points that have already been successfully matched (where $H = 1$) and focus the optimization exclusively on the residual unmatched pairs (where $H = 0$)?
2. In Algorithm 1, are steps 9–12 strictly required to be performed iteratively? I am curious if a simpler Top-K selection could substitute this loop, or if there is a fundamental reason that necessitates a sequential approach.
3. In line 1003, the caption for Appendix C.4 mentions "default hyperparameters" but does not provide their specific values. Specifically, what are the exact values used for $k$ and $n_b$ as referenced in Algorithm 1?  Furthermore, it is unclear whether these settings remained consistent across all quantitative benchmarks.

**Limitations:**

It is essential to discuss the stability and convergence challenges of the optimization process, especially when the matrix $A$ is singular. Additionally, the paper lacks a discussion on the operational bottleneck: how one would efficiently obtain precise alignment supervision in an online setting to make the active acquisition framework practically viable.

**Strengths And Weaknesses:**

Strengths：
1. The framework actively acquires high-quality supervision to bolster alignment accuracy, particularly in scenarios constrained by sparse data labels.
2.  Experiments across diverse alignment tasks show that the proposed method improves alignment performance.



Weaknesses：
1. As shown in the proof, Lemma 4.2 appears to be derived specifically for the entropy-regularized case, yet the paper discusses two other utility functions. Could the authors clarify the analytical formulations or computational procedures for the remaining two cases, as the current lemma does not seem directly applicable to them?
2. In Step 14 of Algorithm 1, the text mentions the update of the alignment supervision matrix $H$. However, the specific procedural details for this update are omitted. To ensure reproducibility, the authors should explicitly define the update rule or the underlying heuristic used in this step.
3. In line 412, the authors state that the impact of the query budget $k$ is detailed in Tables 3–5 and Appendix C.1. However, these specific tables appear to be missing from the provided supplementary material.

---

> ### Author Rebuttal · Authors · 2026-03-30
>
> # Weakness 1: Applicability of Lemma 4.2 to different utility functions
> We would like to respectfully clarify that **Lemma 4.2 is derived independently of specific utility functions** under entropic OT, thus applicable to all utility functions $f(\mathbf{T})$. By Lemma 4.2, the gradient computation only involves $\nabla_\mathbf{T}f$, which **can be computed easily for all three utility functions** we introduce (Table 2).
>
> # Weakness 2: Update of $\mathbf{H}$
> We would like to respectfully clarify that we mention explicitly in **Section 4.2 (line 253)** that, to update the alignment supervision matrix $\mathbf{H}$, we set the corresponding entries of queried alignment pairs to 1. We will update Step 14 in Algorithm 1 to guide readers more effectively.
>
> # Weakness 3: Tables are missing for the query budget $k$
> We would like to respectfully clarify that Table 3-5 are included in the **main body (Section 5)** of our paper, and **Appendix C.1 contains Figure 5-8** of the detailed performance of alignment algorithms vs. query rounds. **All these tables and figures** show that the performance of an alignment method improves as the **query budget $k$ (query rounds)** increases.
>
> # Question 1
>
> ## (1) Does the matrix $H$ serve as a discrete counterpart to the transport plan $T$?
> $\mathbf{H}$ does not serve as a discrete counterpart of the transport map $\mathbf{T}$ since $\mathbf{T}$ is not explicitly enforced to match $\mathbf{H}$. As shown in Eq.(3), $\mathbf{H}$ provides supervision by **penalizing the corresponding entries in the cost matrix** $\mathbf{C}$, thus **implicitly biasing** $\mathbf{T}$ toward pre-known alignments through the OT formulation.
>
> ## (2) Would it be more effective to prune the data points that have already been successfully matched?
> While pruning aligned data points may improve efficiency, it has two main drawbacks. **Firstly**, pruning aligned data points for OT-based alignment algorithms essentially removes all supervision signals, which severely affect performance (Figure 1). **Secondly**, if we keep aligned data points for the alignment method but prune them for AvAtar, this creates **inconsistency in $\mathbf{C}$ and $\mathbf{H}$ used by the optimization of the alignment method and the computation of query impact in AvAtar**, leading to suboptimal query. We empirically verify the above analysis in this [table](https://anonymous.4open.science/r/AvAtar-9278/tables/pruning.png), which shows that pruning degrades the performance of AvAtar.
>
> # Question 2: Are steps 9–12 in Algorithm 1 strictly required to be performed iteratively?
> No, steps 9-12 in Algorithm 1 essentially select the unlabeled data points with the top-k largest posterior query impact.
>
> # Question 3: Hyperparameter settings
> We would like to respectfully clarify that we include hyperparameter settings in **"Reproducibility" in Appendix B.3**, which are **consistent across all datasets, OT-based alignment algorithms, and query methods**. To further improve reproducibility, we include specific values of $k$ and $n_b$ in this [table](https://anonymous.4open.science/r/AvAtar-9278/tables/hyperparameters.png). Please note that although different datasets have different $k$ and $n_b$ since they have different numbers of available ground-truth alignment pairs, **$k$ and $n_b$ are computed consistently as described in Appendix B.3.**
>
> In addition, Appendix C.4 mentions that for **hyperparameters in OT-based alignment algorithms (i.e., PARROT, JOENA, GOT)**, we adopt their default values detailed in their original papers, e.g., $\epsilon$, $\beta$.
>
> # Limitation 1: Lack of discussion on the stability & convergence challenges of the optimization
> We would like to respectfully clarify that **our paper includes both theoretical and empirical convergence results** of the optimization process (i.e., the CG iterations) in AvAtar. For **theoretical analysis**, we show that the CG process is guaranteed to converge to a global optimum with a linear convergence rate, in Lemma 4.3, Appendix A.3, and Lemma 4.6. Please note that we prove in Appendix A.2 that **the coefficient matrix $\mathbf{A}$ in Eq.(6) is always singular**. As shown in Appendix A.3, **the singularity of $\mathbf{A}$ does not affect the convergence of the CG process**. For **empirical evaluation**, Figure 3 shows that the CG process in AvAtar converges empirically with singular $\mathbf{A}$.
>
> # Limitation 2: Lack of discussion on operational bottleneck
> We would like to respectfully clarify that, following standard active alignment works, we focus on designing an **active learning method that selects informative data to query**,  rather than specific data annotation methods. In practice, supervision signals can be obtained online through human annotation or from weak supervision, e.g., metadata, pretrained foundation models, etc. AvAtar minimizes the number of such online queries, therefore reducing the online annotation cost. We will include such a discussion in the revision.

---

> > ### Author Rebuttal · Reviewer_PnUW · 2026-04-02
> >
> > The authors’ explanations and additional experiments have addressed my concerns about the method's practicality. Therefore, I will raise my score.

---

> > > ### Author Response · Authors · 2026-04-06
> > >
> > > Dear reviewer #PnUW,
> > >
> > > Thank you for your acknowledgement of our rebuttal. We are glad to hear that your concerns are addressed. Thank you for raising your score.
> > >
> > > Best,
> > >
> > > Authors

---

### Official Review · Reviewer_rXMp · 2026-03-10

**Soundness:** 3
**Presentation:** 4
**Significance:** 3
**Originality:** 3
**Overall Recommendation:** 5
**Confidence:** 4

**Summary:**

This paper proposes a novel method called AVATAR to address the challenge of sparse and costly supervision in OT-based alignment tasks. The method quantifies the impact of candidate samples on global alignment results through gradient propagation in entropy-regularized OT. It achieves linear complexity via the adjoint-state method and conjugate gradient solver. Theoretical analysis establishes its correctness and convergence. Extensive experiments show the effectiveness of the proposed method.

**Compliance With Llm Reviewing Policy:**

Affirmed.

**Final Justification:**

My concerns and questions are resolved, and i update the score to 5.

**Key Questions For Authors:**

Question
1. How sensitive is AVATAR to the entropy regularization parameter ε? Does it degrade for small ε (closer to unregularized OT) or large ε (over-smoothing)?
2. In experiments, what are the specific quantitative gains over baselines (e.g., exact % improvements per task/dataset)? Are there cases where AVATAR underperforms, and why?
3. Could AVATAR be extended to query target objects (Y) instead of only source (X), or to multi-object queries?
4. The consistency utility uses graph Laplacians; how does it handle non-graph data in CDA tasks, or could it be generalized?

**Limitations:**

To enhance the completeness of the paper, I suggest the authors add a brief discussion about the limitations of their work.

**Strengths And Weaknesses:**

Strength
1. The paper introduces a timely and important problem of active learning specifically tailored for optimal transport (OT)-based alignment methods, which is underexplored. AVATAR provides a principled framework that leverages gradient propagation through entropy-regularized OT to quantify query impact, addressing key limitations in prior active alignment works (e.g., lack of integration with OT components like costs and constraints).
2. The use of the adjoint-state method to reformulate differentiation through OT into a linear system solvable by conjugate gradient (CG) with linear complexity is clever and efficient. This enables scalability for large-scale alignment problems. The utility functions (L2, entropy, consistency) are well-motivated and adaptable to different tasks.
3. The method is task-agnostic, demonstrated across diverse alignment tasks (network alignment, image-text retrieval, grounding) with 8 datasets, 4 OT methods, and 9 baselines. Experiments show consistent improvements (e.g., up to 15% in performance with better supervision), and the analysis of complexity/convergence is thorough.
4. The problem formulation is unified and clear. Proofs (e.g., for Lemma 4.2, 4.3) and Algorithm 1 provide solid foundations, with guarantees on convergence and efficiency.

Weakness
1. While the utility functions are effective empirically, their selection (e.g., why L2 and entropy for all tasks, consistency only for NA) lacks deeper theoretical motivation or ablation studies showing sensitivity to alternatives. It's unclear if these are optimal or if task-specific tuning is needed.
2. The method relies heavily on entropy-regularized OT and assumes a perfect oracle with binary supervision (H matrix). It may not extend easily to unbalanced OT, Gromov-Wasserstein, or noisy/real-world oracles. The paper mentions unsupervised/weakly supervised settings but doesn't address fully supervised scenarios or integration with deep OT models (e.g., neural OT solvers).
3. Comparisons are strong, but details on runtime (e.g., actual wall-clock times for large n/m) are missing, despite complexity claims. Ablations on hyperparameters (e.g., ε, β, batch size nb) or failure cases (e.g., when gradients are uninformative) could strengthen results.
4. While limitations of prior active alignment are highlighted, deeper connections to general active learning (e.g., expected gradient length in AL literature) or recent OT advances (e.g., differentiable OT in ML) could be expanded.

---

> ### Author Rebuttal · Authors · 2026-03-31
>
> # Weakness 1: Utility functions lack theoretical insight
> We would like to respectfully clarify that our choices of $f_{L_2}$ and $f_{\text{entropy}}$ are guided by their simplicity and task-agnostic nature, which encourage deterministic alignment results desirable for most tasks that identify hard alignments [1]. For $f_{\text{consist}}$, it encourages the consistency principle in NA, and is particularly effective on plain networks (Table 3).
>
> # Weakness 2: Reliance on entropic OT and perfect oracles / Supervised settings and neural OT are not discussed
> We thank the reviewer for this important point. We would like to clarify that AvAtar is not restricted to balanced entropic OT. Specifically, for **unbalanced entropic OT (UOT)** solvable by the generalized Sinkhorn algorithm, AvAtar can be easily extended to UOT by differentiating through its non-constrained objective; for **Gromov-Wasserstein** variants, [1] shows that it can be decomposed into a sequence of entropic OT subproblems, making AvAtar directly applicable by differentiating through this iterative OT solver.
>
> Secondly, **AvAtar does not rely on oracles which provide deterministic supervision**. Instead, AvAtar can be easily extended to soft, non-deterministic supervision, where a query may return multiple candidate matches (or multiple queries return one candidate match), affecting multiple entries within a row (column) of $\mathbf{H}$. In this case, $\mathbf{H}$ becomes a soft supervision matrix with continuous values, making AvAtar directly applicable.
>
> Thirdly, we would like to respectfully clarify that AvAtar is designed for **standard active learning settings**, where supervision is limited and costly to obtain.
>
> Finally, we agree that extending AvAtar to neural OT could be an interesting future direction, and will include it in the limitation & future work section in our revision.
>
> # Weakness 3: Runtime & hyperparameter sensitivity study
> We would like to respectfully clarify that we include detailed wall-clock time (total query time) of AvAtar compared to other active alignment methods in the **efficiency (Section 5.3) and scalability (Appendix C.2) study**.
>
> For hyperparameters, we would like to clarify that **$\epsilon$ and $\beta$ are set by a specific OT-based alignment algorithm instead of hyperparameters in AvAtar**, whose values can be found in their original paper. For $n_b$, we include a hyperparameter study in Section 5.1.1, which shows that the performance of AvAtar remains stable under different choices of $n_b$.
>
> We further study the sensitivity of AvAtar to $\epsilon$ in this [table](https://anonymous.4open.science/r/AvAtar-9278/tables/epsilon.png), and $\beta$ in this [table](https://anonymous.4open.science/r/AvAtar-9278/tables/beta.png), which shows that AvAtar consistently achieves the best performance compared to baseline query methods.
>
> # Weakness 4: Lack of discussion on connections to general active learning and recent OT advances
> We thank the reviewer for this valuable suggestion. The query impact formulation in AvAtar is closely related to general gradient-based active learning methods, e.g., expected gradient length (EGL), and can be viewed as an extension of EGL to OT-based alignment. In addition, our theoretical results provide an efficient way to differentiate through the entropic OT formulation, contributing to differentiable OT in general ML problems.
>
> # Question 1: Sensitivity of AvAtar to $\epsilon$
> We would like to respectfully refer the reviewer to our response to Weakness 3, which includes an additional sensitivity study on $\epsilon$.
>
> # Question 2: Quantitative gains & failure cases
> We would like to respectfully clarify that we include quantitative gains (in %) over baselines in our analysis of the benchmarking results (Section 5.2). We will add a column in Table 3-5 to further highlight detailed gains in %.
>
> In our experiments, we do not notice any specific cases where AvAtar fails or consistently underperforms. Comprehensive parameter studies over $\epsilon$, $\beta$, $n_b$ show that AvAtar is consistently effective across different choices of parameters, implying that the gradient remains informative in characterizing the most informative candidates.
>
> # Question 3: Extension to multi-object queries
> We would like to respectfully refer the reviewer to our response to Weakness 2, which detailed such an extension.
>
> # Question 4: Applicability of $f_{\text{consist}}$ to CDA
> We would like to respectfully clarify that $f_{\text{consist}}$ is designed specifically for NA tasks. That said, this idea can be extended to CDA tasks by constructing a similarity graph from CDA data [2], making $f_{\text{consist}}$ directly applicable.
>
>
> # Reference
> [1] Yu, Qi, et al. "Joint optimal transport and embedding for network alignment." Proceedings of the ACM on Web Conference 2025. 2025.
>
> [2] Chen, Liqun, et al. "Graph optimal transport for cross-domain alignment." International Conference on Machine Learning. PMLR, 2020.

---

> > ### Author Rebuttal · Reviewer_rXMp · 2026-04-02
> >
> > My concerns and questions are resolved, and i will consider raising my score.

---

> > > ### Author Response · Authors · 2026-04-06
> > >
> > > Dear reviewer #rXMp,
> > >
> > > Thank you for your acknowledgement of our rebuttal. We are glad to hear that your concerns and questions are resolved, and you are considering raising your score. We would be more than happy to address any further questions or concerns regarding our paper.
> > >
> > > Best,
> > >
> > > Authors
> > >
> > > --------
> > >
> > > Dear reviewer #rXMp,
> > >
> > > Thank you again for your thoughtful review and your engagement during the rebuttal period. Since you mentioned that you will consider raising your score, we would like to kindly follow up on your decision, and would be happy to address any additional questions or concerns regarding our paper.
> > >
> > > We sincerely appreciate your time and consideration.
> > >
> > > Best,
> > >
> > > Authors
> > >
> > > ----
> > >
> > > Dear reviewer #rXMp,
> > >
> > > Thank you again for raising your score.
> > >
> > > Best,
> > >
> > > Authors

---

### Official Review · Reviewer_FRWK · 2026-03-10

**Soundness:** 3
**Presentation:** 3
**Significance:** 3
**Originality:** 3
**Overall Recommendation:** 4
**Confidence:** 4

**Summary:**

The paper proposes an active learning framework for Optimal Transport - based alignment methods. The proposed active learning framework is based on gradient activations of certain utility functions with respect to the (currently available, at the current active learning round) supervision matrix; in turn, the gradient activations are obtained with Conjugate Gradient iterations of a specifically designed linear system. The methodology is benchmarked on different alignment tasks: Network Alignment, Image-text retrieval and Image-text grounding.

**Compliance With Llm Reviewing Policy:**

Affirmed.

**Key Questions For Authors:**

See above

**Limitations:**

Limitations are not discussed. Yet I think authors need to discuss the limitations of their method

**Strengths And Weaknesses:**

**Strengths**

The paper is clear and well-written, for me the structure of the paper is very good. The authors try to formally define and state the (mathematical) problems and setups they solve (e.g., what is OT-based Alignment, what is Active OT-based alignment). The experimental validation is rather extensive, the obtained quality comparisons are promising.

**Weaknesses**

I do not see serious weaknesses for the paper. Still, I have some questions and comments for the authors.

1. Line 144: I think, cost matrix $C$ should contain non-negative values, otherwise the OT-based alignment setup, eq. (3) looks strange.

2. The authors need to introduce the definitions of the used evaluation metrics, MRR and Recall@1 in the Experiments section or somehow in the Appendix.

3. The authors need to introduce the considered problems more clearly. In particular, the authors say that for Image Text Retrieval problem they use CIFAR-10-C and ImageNet-C datasets. But how particularly the Image Text Retrieval problem looks like in the case of these datasets? Same question for Image-Text Grounding.

4. Section 5.3. What is query time? How is it defined?

5. I think an important addition that could improve the manuscript would be a comparison with non-OT alignment methods. In particular, Tables 3, 4 and 5 consider OT-based alignment methods alongside non-OT algorithms intended for active alignment (apart from the authors' proposed Avatar method). This naturally raises the question of how OT-based alignment with an OT-specific active learning approach works compared with non-OT-based alignment with a non-OT-specific active learning approach. The paper would benefit from such a comparison.

---

> ### Author Rebuttal · Authors · 2026-03-30
>
> # Question 1: Cost matrix $\mathbf{C}$ should contain non-negative values
> We thank the reviewer for pointing this out, and will update $\mathbf{C}\in\mathbb{R}^{n\times m}_{\geq 0}$ in our notations.
>
> # Question 2: Definitions of MRR and Recall@1 are missing
> We thank the reviewer for this helpful suggestion. While these two metrics are standard in NA and CDA, we agree that defining them explicitly improves the clarity of our paper. For **mean reciprocal rank (MRR)**, it is defined as the average reciprocal of the rank at which the correct alignment appears in the candidate list, i.e., $\text{MRR}=\frac{1}{N}\sum_{i=1}^{N}\frac{1}{rank_i}$, where $N$ is the size of source object set, and $rank_i$ is the rank of the correct alignment for the $i$-th object in the source set; for **Recall@1**, it is defined as the proportion of source object whose correct alignment is ranked 1st by an alignment method, i.e., Recall@1$=\frac{1}{N}\sum_{i=1}^{N}\mathbb{1}(rank_i=1)$, where $\mathbb{1}\{\}$ denotes the indicator function. We will include these definitions in the Appendix.
>
> # Question 3: Definitions of cross-domain alignment on specific datasets
> We thank the reviewer for this helpful suggestion. Follwing standard cross-domain alignment settings [1], for **image-text retrieval on CIFAR-10-C and ImageNet-C**, we construct image-text pairs by associating each images with the corresponding textual description based on its class label, and treat retrieval as matching images to their corresponding text embeddings; for **image-text grounding on COCO and Flickr30K Entities**, which contains explicit fine-grained correspondence between phrases in a sentence and objects (or regions) in an image, we treat grounding as an matching problem between phrases and objects for an image.
>
> # Question 4: Definition of query time
> We thank the reviewer for this clarification question. We define the total query time in Section 5.3 as the accumulated runtime of different active alignment methods under a fixed query budget $k$ and query batch size $n_b$. Formally, $t^{(k, n_b)}=\sum_{i=1}^{N}t_i^{(k, n_b)}$, where $t_i^{(k, n_b)}$ is time required to select query candidates at the i-th round, after the completion of the alignment algorithm. $N$ denotes the total number of query rounds. **$k$, $n_b$, and $N$ are consistent for all active alignment methods for fair comparison**.
>
> # Question 5: Comparision with non-OT-based alignment algorithms + non-OT-based active learning approach
> We would like to first clarify that the alignment algorithms adopted in Figure 3-5, i.e., PARROT, JOENA, and GOT, are all OT-based methods. The active learning baselines are designed for non-OT-based methods except for the proposed AvAtar, which is the first OT-specific active learning approach to our best knowledge.
>
> We agree that such a comparision would improve the comprehensiveness of our paper. To further demonstrate the power of OT-based alignment algorithm (e.g., PARROT) equipped with OT-specific active learning approach (AvAtar), we compare its performance with a non-OT-based alignment algorithm (FINAL [2]) with a non-OT-specific active learning approach (Attent [3]). The results are shown in this [table](https://anonymous.4open.science/r/AvAtar-9278/tables/non-OT.png), which shows that **1)** PARROT + AvAtar consistently outperforms FINAL + Attent, demonstrating the power of OT-based alignment, **2)** the performance gap increases as the query round (budget) increases, showing the superiority of AvAtar in boosting the performance of OT-based alignment methods.
>
> # Reference
> [1] Chen, Liqun, et al. "Graph optimal transport for cross-domain alignment." International Conference on Machine Learning. PMLR, 2020.
>
> [2] Zhang, Si, and Hanghang Tong. "Final: Fast attributed network alignment." Proceedings of the 22nd ACM SIGKDD international conference on knowledge discovery and data mining. 2016.
>
> [3] Zhou, Qinghai, et al. "Attent: Active attributed network alignment." Proceedings of the Web Conference 2021. 2021.

---

> > ### Author Rebuttal · Reviewer_FRWK · 2026-04-03
> >
> > I thank the authors for the answer. I am keeping my positive score.

---

> > > ### Author Response · Authors · 2026-04-06
> > >
> > > Dear reviewer #FRWK,
> > >
> > > Thank you for your acknowledgement of our rebuttal and your positive recommendation of our paper.
> > >
> > > Best,
> > >
> > > Authors

---

### Official Review · Reviewer_7z5G · 2026-03-12

**Soundness:** 3
**Presentation:** 2
**Significance:** 3
**Originality:** 3
**Overall Recommendation:** 5
**Confidence:** 4

**Summary:**

The challenge of object alignment is widespread in machine learning, encouraging the search for robust methods. In this sense, optimal transport (OT), notably the Kantorovich problem, represents a mathematically rigorous framework.
Motivated specifically by network (NA) and cross-domain (CDA) alignment, this work introduces a novel active learning framework for the OT alignment problem called AVATAR. The entropic OT formulation is generalized to account for a finite budget ($k$) of supervisions, provided by an oracle. The source samples $x_i \in \mathcal{X}$ to be supervised are chosen by quantifying their posterior gradient-based impact on the global alignment, which is computed via the adjoint-state and conjugate gradient methods. \
Through theoretical complexity and convergence analysis, as well as benchmarking results in NA and CDA contexts, the authors demonstrate the interest of AVATAR with respect to active OT-baselines.

**Compliance With Llm Reviewing Policy:**

Affirmed.

**Final Justification:**

The author’s rebuttal has successfully clarified my concerns, presenting additional experiments, such as the drift study between queries, which, in my view, shed further light on the research interest. Nevertheless, I would suggest adding a section in the appendix outlining the limitations of the approach (or in general active OT-based approaches). In my opinion, the paper deserves to be accepted.

**Key Questions For Authors:**

1) Eq (3) introduces a penalizing factor $\beta$ that controls the strength of the supervisions. However, you never detail settings of this parameter, nor discuss its influence on the AVATAR framework. Could you provide an ablation study or a principled heuristic for setting this parameter across different tasks?

2) The problem definition assumes an oracle provides "the correct alignment" $y_t$ for a source object $x_s$. How does the proposed framework empirically handle scenarios where the oracle provides non-deterministic (one-to-many) soft correspondences?

3) It is know that entropic OT (Sinkhorn) formulation is strictly convex, thus it yields a unique optimal transport plan $T^\ast$ for a given cost matrix $C$. AVATAR iteratively updates the supervision matrix $H$ over the batch queries and actively modifies the penalized cost $\tilde{C}$, forcing OT computation to converge to a sequence of new unique optimal plans. Have you considered monitoring how far the supervised plans "drift" from the initial solution $T^{(0)}$, as the active learning loop progresses? Including such an analysis would provide deep insights into the corrective benefit of the oracle.

**Limitations:**

Yes

**Strengths And Weaknesses:**

AVATAR tackles the challenge of active alignment within the OT framework and specifically addresses the high cost of acquiring quality supervision. The key methodological strength lies in the efficient approach to computing the informativeness of source samples. As authors highlight, differentiating through the constrained OT optimization problem normally requires explicitly forming and inverting a massive, computationally intractable Jacobian matrix. AVATAR bypasses this by reformulating the gradient computation into an adjoint linear system that is efficiently solvable via the conjugate-gradient (CG) method. Theoretical guarantees are provided, specifically proving linear time complexity and a linear convergence rate for the CG method, ensuring the active learning loop remains scalable.

The paper is generally well-written, however, $T$ is consistently referred to as the OT "transport map". In the OT framework, $T\in\mathbb{R}^{n\times m}$ is referred to as the "transport plan". The OT transport map is usually associated to the Monge map $T:\mathbb{R}^d\to\mathbb{R}^d$ such that $T​_\sharp\mu=\nu$ (following the push-forward operator).

Furthermore, from a first reading, one might think that AVATAR is applicable to all types of alignment tasks. However, I believe that the explicit claim of being solution of (_Limitation #3_) is not entirely fair. The experimental section focuses specifically on NA and CDA, tasks in which the cost matrix $C$ usually is noisy, and where a deterministic supervision of an oracle proves crucial. In scenarios where data creates explicit geometries (e.g., 2D/3D shape/point cloud matching), standard unsupervised OT (Eq (1) and (2)) techniques are highly efficient. In such settings, AVATAR framework and and the reliance on an oracle to identify deterministic point-to-point matches would likely be impractical. I would suggest to clearly scope in a _Remark_ the limitations of active OT-based approaches for completeness.

## Minor remarks:
- In the caption of Figure 1 I suggest clarifying the phrasing to "_active_ OT-based alignment methods" for better context
- The query batch size is denoted as $n_b$​ in Algorithm 1, but it is later referred to as $b$ in the theoretical results of Section 4.3
- Algorithm 1, Line 11: change $x_i$ with $x^\ast$ (matching the variable assigned in Line 10)

---

> ### Author Rebuttal · Authors · 2026-03-31
>
> # Weakness 1: T should be referred to as the transport plan
> We thank the reviewer for this helpful comment. We acknowledge that $\mathbf{T}$ in our settings should be more precisely referred to as the "transport plan", and we will make sure to align with standard OT terminology in the revision of our paper.
>
> # Weakness 2: Limitation of active OT-based approaches and AvAtar
> We thank the reviewer for this insightful comment. We acknowledge that the effectiveness of unsupervised OT in 2D/3D point cloud matching may imply robust cost functions, thus limiting the **relative benefit** of AvAtar. However, this does not render AvAtar inapplicable, especially under **imperfect cost design** due to outlier points and noisy features [1].
>
> Importantly, **AvAtar does not rely on oracles that provide strictly deterministic supervision**. Instead, AvAtar can be easily extended to soft, non-deterministic supervision, where a query may return multiple candidate matches, affecting multiple entries within a row of $\mathbf{H}$. In this case, $\mathbf{H}$ becomes a soft supervision matrix with continuous values, instead of a binary matrix that encodes hard supervision. In this case, AvAtar remains directly applicable without modification to any other components.
>
> Overall, AvAtar is most effective under a noisy cost matrix $\mathbf{C}$, and can be easily extended to scenarios where the the oracle provides non-deterministic correspondences. The benefit of AvAtar may be partially reduced but not eliminated in some geometric matching tasks, where the cost functions are relatively robust. We will clarify this point and discuss such limitations in the revision.
>
> # Weakness 3: Minor Remarks
> We thank the reviewer for these fine-grained observations and will correct these notations in the revision of our paper.
>
> # Question 1: Penalizing factor $\beta$
> We would like to respectfully clarify that $\beta$ is set by a specific OT-based alignment algorithm (i.e., PARROT, JOENA, GOT) rather than a hyperparameter in AvAtar, and its specific value can be found in the corresponding paper and implementation. To further improve our paper, we study the sensitivity of AvAtar to $\beta$ in this [table](https://anonymous.4open.science/r/AvAtar-9278/tables/beta.png), which shows that AvAtar consistently achieves the best performance compared to other query methods under different $\beta$.
>
> # Question 2: Non-deterministic oracles
> We would like to respectfully refer the reviewer to our response to Weakness 2, which includes how AvAtar handles oracles that provide soft, non-deterministic correspondences.
>
> # Question 3: Drifts of $\mathbf{T}$
> We thank the reviewer for this helpful suggestion. We agree that tracking how the transport plan $\mathbf{T}$ drifts from $\mathbf{T}^{(0)}$ could further improve the interpretability regarding the corrective benefit of the oracle. We include an additional study on the drift of $\mathbf{T}^{(i)}$ from $\mathbf{T}^{(0)}$ in the $i$-th query round, measured by the frobenius norm of their difference (i.e., $\|\|\mathbf{T}^{(i)}-\mathbf{T}^{(0)}\|\|_{\text{F}}$). The results are shown in [figure](https://anonymous.4open.science/r/AvAtar-9278/figures/T_drift.png), which shows that **1)** $\mathbf{T}^{(i)}$ gradually drifts from $\mathbf{T}^{(0)}$ with increasing level of supervision along query rounds, and **2)** the drift of $\mathbf{T}$ positively correlates with the performance improvement of OT-based alignment methods, suggesting that the acquired supervision from the orcale gradually corrects $\mathbf{T}$ to infer more accurate alignment. We will include this study in the revision of our paper.
>
>
> # Reference
> [1] Qin, Zheng, et al. "Geometric transformer for fast and robust point cloud registration." Proceedings of the IEEE/CVF conference on computer vision and pattern recognition. 2022.

---

> > ### Author Rebuttal · Reviewer_7z5G · 2026-04-03
> >
> > I would like to thank the authors for their clarification and I will update my rate.

---

> > > ### Author Response · Authors · 2026-04-06
> > >
> > > Dear reviewer #7z5G,
> > >
> > > Thank you for your acknowledgement of our rebuttal. We are glad to hear that you will update your score, and we would be more than happy to address any further questions or concerns regarding our paper.
> > >
> > > Best,
> > >
> > > Authors
> > >
> > > --------------
> > >
> > > Dear reviewer #7z5G,
> > >
> > > Thank you again for your thoughtful review and your engagement during the rebuttal period. Since you mentioned that you will update your score but have not yet done so, we would like to kindly follow up to see if you have additional questions or concerns regarding our paper, and we would be happy to address them.
> > >
> > > We sincerely appreciate your time and consideration.
> > >
> > > Best,
> > >
> > > Authors

---

### Decision · Program_Chairs · 2026-04-30

**Decision:**

Accept (regular)

**Comment:**

In this paper the authors propose an active learning strategy for aligning two
distributions with optimal transport. The method is based on a strategy to
select the points to be labeled that are the most informative for the OT plan.
The method is shown to be more efficient than random sampling and other active
learning strategies on synthetic and real data.

The reviewers found the paper interesting but had a few concerns about the
details in numerical experiments, sensitivity to parameters and comparison with
other non-OT active learning strategies. The authors provided a detailed
response to the reviews with new experiments and clarifications that were
appreciated by all reviewers. They all stated that their concerns were fully resolved. I agree that the paper is interesting and that the method is novel so I recommend an acceptance but I expect the authors to include in the final version all the clarifications and new experiments that they did in the response.